# DNA metabarcoding-based diet survey for the Eurasian otter (*Lutra lutra*): Development of a Eurasian otter-specific blocking oligonucleotide for 12S rRNA gene sequencing for vertebrates

**Priyanka Kumari**[1,2], **Ke Dong**[1,2¤], **Kyung Yeon Eo**[3], **Woo-Shin Lee**[4], **Junpei Kimura**[5], **Naomichi Yamamoto**[1,2]*

**1** Department of Environmental Health Sciences, Graduate School of Public Health, Seoul National University, Seoul, South Korea, **2** Institute of Health and Environment, Graduate School of Public Health, Seoul National University, Seoul, South Korea, **3** Conservation and Research Center, Seoul Zoo, Gwacheon, South Korea, **4** Department of Forest Sciences, College of Agriculture and Life Science, Seoul National University, Seoul, South Korea, **5** College of Veterinary Medicine, Seoul National University, Seoul, South Korea

¤ Current address: Life Science Major, College of Convergence and Integrated Science, Kyonggi University, Gyeonggi-do, South Korea

* nyamamoto@snu.ac.

**Data Availability Statement:** All relevant data are available within the manuscript and Supporting

## Abstract

The Eurasian otter (*Lutra lutra*) is an endangered species for which diet analyses are needed as part of its conservation efforts. Eurasian otters feed on vertebrates, such as fishes, and invertebrates, such as crustaceans, but their detailed taxonomies are not fully understood in part due to limited resolving power of traditional morphological identification methods. Here, we used high-throughput sequencing (HTS)-based DNA metabarcoding approaches to analyze diet profiles of Eurasian otters inhabiting a marshy estuary area in Korea. We investigated their diet profiles based on spraint sampling followed by DNA metabarcoding analyses targeting 12S rRNA gene region for vertebrates, 16S rRNA gene region for invertebrates, and cytochrome *c* oxidase 1 (COI) gene region for fishes. For the vertebrate analysis, a blocking oligonucleotide (OBS1) was designed to suppress amplification of DNA fragments derived from the otters. The 12S rRNA gene sequencing assay detected species belonging to fishes (95%) and amphibians (3.3%). Fishes detected by 12S rRNA gene sequencing included crucian carp (*Carassius auratus*), mullets (*Mugil* spp.), bluegill (*Lepomis macrochirus*), and northern snakehead (*Channa argus*), which were also detected by COI gene sequencing. Among invertebrates, mud flat crabs (*Helicana* spp.) and shrimps (*Palaemon* spp.) were abundant. The designed blocking oligonucleotide OBS1 effectively inhibited amplification of the otter's DNA, with only up to 0.21% of vertebrate sequence reads assigned to the otter. This study demonstrated that HTS-based DNA metabarcoding methods were useful to provide in-depth information regarding diet profiles of the otters at our sampling site. By using HTS-based DNA metabarcoding approaches, future research will explore detailed taxonomies of their diets across locations and seasons.

Information files. Raw sequence data are available at NCBI under the BioProject accession numbers PRJNA497180, PRJNA548047, PRJNA548048 and PRJNA565647.

**Funding:** This work was supported by Seoul National University Research Grant in 2018 (W-SL, JK, and NY). The funders had no role in study design, data collection and analysis, decision to publish, or preparation of the manuscript.

**Competing interests:** The authors have declared that no competing interests exist.

## Introduction

Eurasian otters (*Lutra lutra*) are widely distributed from Europe to Asia, including South Korea. Due to its declining population, however, the Eurasian otter is listed as a near threatened species by the International Union for Conservation of Nature Red List [1] and as a class I endangered species by the Ministry of Environment of Korea [2]. Eurasian otters feed on fishes, amphibians, crabs, birds, and insects [3, 4], and are at risk of intake of polluted diets, which may lead to a further decline in their population [5]. Therefore, it is critical to survey their diets, including potentially contaminated preys, as an effort for their conservation.

The otters' feces (spraints) are useful for their diet analyses [3, 4, 6] and individual identification [7, 8]. Morphological observation of undigested food remains, such as bones, shells, and hair, in spraints provides information regarding the otters' diets [3, 6]. Previous studies reported fishes as the otters' main preys followed by amphibians, crabs, birds, and insects [3, 4]. Nonetheless, knowledge is limited regarding their detailed taxonomies in part due to limited resolving power of traditional morphological methods [9]. Meanwhile, Hong et al. [10] used a Sanger sequencing-based approach to identify vertebrate species for each individual bone remain isolated from spraints. However, this approach is laborious and requires technical expertise that limits the capacity of information generation.

Emerging high-throughput sequencing (HTS)-based DNA metabarcoding methods can circumvent such difficulties, and are used for diet analyses of wild animals, such as leopard cat in Pakistan [11], puma, jaguar, ocelot, and crab-eating fox in Venezuela [12], and Iberian lynx in Spain [13]. Due to its large capacity of information generation, the HTS-based methods can improve taxonomic resolution in greater details with limited quantities of fecal analytes [11, 14, 15]. However, the DNA metabarcoding-based diet analyses can potentially suffer from amplification of DNA fragments derived from a predator rather than preys, resulting in the reduced sensitivity in rare prey detection [11]. The most effective solution to overcome this problem is to use blocking oligonucleotides that suppress amplification of predator DNA [16]. We expect that a similar approach would be useful for diet analyses of the Eurasian otters inhabiting Korea.

Here, we investigated vertebrate and invertebrate diet profiles of the Eurasian otters inhabiting a marshy estuary area of South Korea based on spraint sampling followed by HTS-based DNA metabarcoding analyses targeting 12S rRNA gene region for vertebrates, 16S rRNA gene region for invertebrates, and cytochrome *c* oxidase 1 (COI) gene for fishes. We used previously reported universal primers for identification of vertebrates [17], invertebrates [15], and fishes [18]. In particular, we aimed to develop a blocking oligonucleotide that inhibits amplification of the otter's DNA when preparing libraries for the vertebrate-specific 12S rRNA gene sequencing assay. The performance of the designed blocking oligonucleotide was assessed in this study.

## Materials and methods

### Study area and sample collection

A total of seven spraints (S1 Fig) were collected on June 6, 2017 in and with permission from the Ansan Reed Marshy Park (37°16'22.6"N 126°50'24.7"E) in an estuary area in Ansan-si in Gyeonggi-do in South Korea (Fig 1). The park was situated along a branch stream flowing into Sihwaho Lake, a regulating brackish lake separated from the Yellow Sea by a seawall. About 10 g of each spraint sample was collected using a sterile wooden spatula into a 50 ml tube. The collected samples were kept on ice packs and transported to the laboratory on the same day of sample collection. The samples were kept at –80°C until DNA extraction.

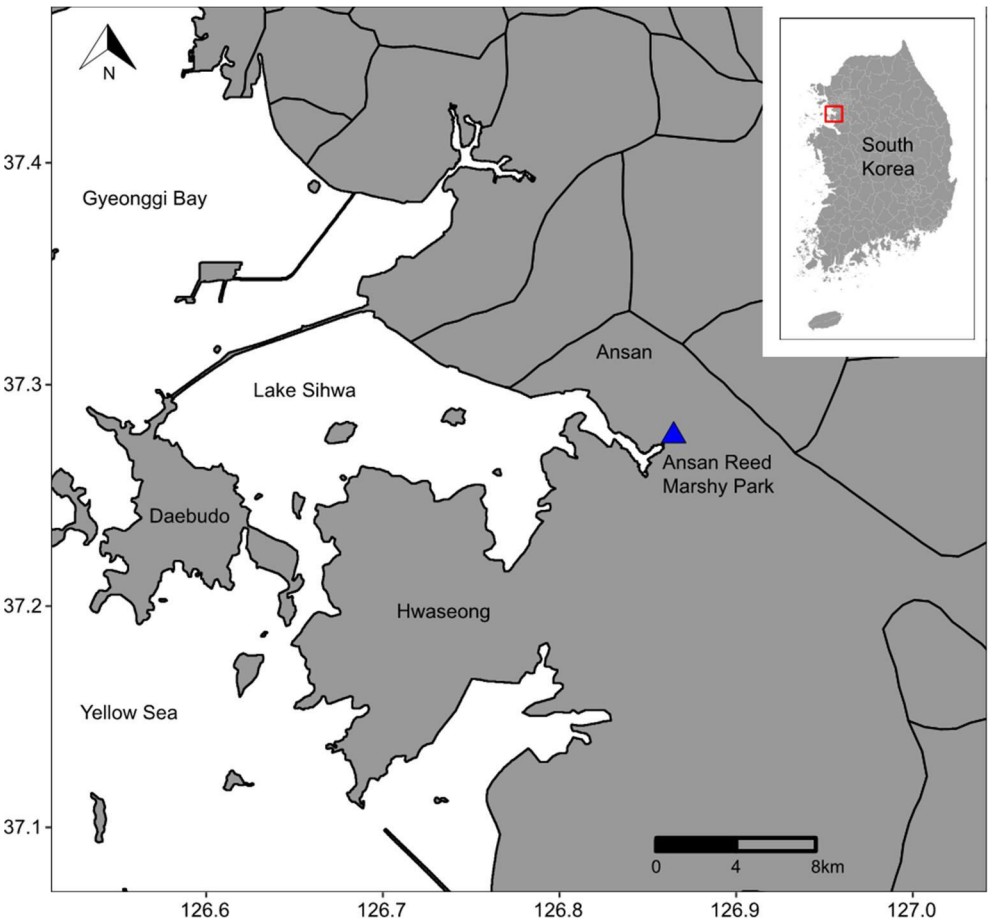

**Fig 1. Sampling location for spraints in this study.**

## DNA extraction and sample confirmation

DNA was extracted from spraints using the PowerMax® Soil DNA Isolation Kit (Mobio Laboratory, Inc., Carlsbad, CA, USA) with modifications [19]. About 5 ml of ultra-pure water was added to each sample, and manually homogenized in the 50 ml tube by a sterile wooden spatula. About 0.2 g of each homogenized sample, including visible bone remains and scales, was transferred into a 2 ml tube containing the Mobio's Power Beads and solution (750 ml) with additional 0.1 mm diameter glass beads (300 mg) and 0.5 mm diameter glass beads (100 mg) [20]. The samples were homogenized for 3 min by a bead beater (BioSpec Products, OK, USA). After homogenization, DNA was extracted and eluted into 50 μl of TE (10 mM Tris-HCl, 1 mM EDTA, pH = 8.0) according to the kit's protocol. For the 12S and 16S rRNA gene analyses, three different subsamples (each with 0.2 g) of each homogenized sample in the 50 ml tube were transferred into three different 2 ml tubes for DNA extraction and purification, and recombined for elution into a tube by 50 μl of TE. For the COI analysis, only one part was used for DNA extraction and elution into a tube by 50 μl of TE. The extracted DNA concentrations ranged from 16.6 to 38.9 ng μl$^{-1}$ for the 12S and 16S rRNA gene analyses, and from 2.5 to 10.6 ng μl$^{-1}$ for the COI analysis (S1 Table). Each collected sample was analyzed for its identity by a Eurasian otter-specific PCR assay with primers LutcytF and LutcytR targeting the partial cytochrome *b* sequence [7]. All of the samples were confirmed to be of spraints (S2 Fig).

**Table 1. Blocking oligonucleotide OBS1 targeting the 12S rRNA gene region of the Eurasian otter.** The sequence of the blocking oligonucleotide OBS1 is compared with the sequences of the 12S rRNA gene region of relative species and potential preys of Eurasian otters. The dot (.) represents the same type of nucleotides as that of the blocking oligonucleotide OBS1 and the dash (-) represents the gap in the sequence alignment.

| Accession no. | Type or species name | Sequence (5'-3') | | | | | | | | | | | | | | | | | | | | | | | | | | | | | | | | | | | | | | | | | | | | |
|---|---|---|---|---|---|---|---|---|---|---|---|---|---|---|---|---|---|---|---|---|---|---|---|---|---|---|---|---|---|---|---|---|---|---|---|---|---|---|---|---|---|---|---|---|---|---|
| OBS1 | Blocking oligonucleotide | C | T | A | T | G | C | T | C | A | G | C | C | C | T | A | C | A | A | A | C | A | T | A | T | A | G | C | T | T | A | C | A | T | A | A | C | A | A | A | A | C | T | A | T | C | T | G | C |
| EF672696 | *Lutra lutra* | . | . | . | . | . | . | . | . | . | . | . | . | . | . | . | . | . | . | . | . | . | . | . | . | . | . | . | . | . | . | . | . | . | . | . | . | . | . | . | . | . | . | . | . | . | . | . | . |
| KY117556 | *Lutra sumatrana* | . | . | . | . | . | . | . | . | . | . | . | . | . | . | . | . | . | . | . | . | . | . | . | . | . | . | . | . | . | . | . | . | . | . | . | . | . | . | C | . | . | . | . | . | . | . | . | . |
| AB119064 | *Mustela altaica* | . | . | . | . | . | . | . | C | . | . | . | . | . | . | . | . | A | . | . | C | C | . | . | . | . | . | . | . | . | . | . | . | . | . | T | . | . | . | . | . | . | . | . | . | . | . | . | . |
| AB291075 | *Meles anakuma* | . | . | . | . | T | . | . | . | . | . | . | . | . | G | . | A | . | T | . | C | C | . | . | . | G | . | . | C | . | . | . | . | T | . | . | . | . | . | . | . | . | . | . | . | . | . | . | . |
| AB119065 | *Mustela nivalis* | . | . | . | . | . | . | . | . | . | A | . | . | . | . | . | A | T | . | . | C | C | . | . | . | C | . | . | . | . | . | . | . | T | . | . | . | . | . | . | T | . | . | . | . | . | . | . | . |
| AB291077 | *Enhydra lutris* | . | . | . | . | . | . | . | . | . | G | . | . | . | . | . | A | T | . | . | A | . | . | . | . | C | . | . | . | . | . | . | . | T | . | . | . | . | . | . | . | . | . | . | . | . | . | . | . |
| AP006041 | *Channa argus* | . | . | . | T | . | . | . | . | . | T | . | . | . | . | . | . | . | . | A | C | C | . | . | . | T | A | C | T | A | C | . | . | . | . | G | . | . | . | . | E | T | C | . | . | . | . | . | . |
| AP002930 | *Mugil cephalus* | C | . | . | . | . | . | . | . | . | T | . | . | . | T | . | A | T | . | . | . | . | . | . | . | T | C | A | C | . | . | C | C | - | . | . | . | T | . | . | . | . | C | . | . | . | . | . | . |
| AB006953 | *Carassius auratus* | . | . | . | G | . | . | . | . | . | G | . | . | . | T | . | A | . | . | C | T | A | . | . | . | T | A | C | . | A | T | . | . | . | I | . | G | A | . | G | . | . | C | . | . | . | . | . | . |
| KM590550 | *Rana coreana* | . | . | . | G | . | . | A | T | T | . | A | C | T | T | . | A | C | T | T | A | C | . | A | C | T | C | C | . | G | A | C | C | C | . | . | G | . | . | . | . | . | . | . | . | . | . | . | . |
| AF150004 | *Lepomis macrochirus* | . | . | . | C | T | . | . | . | . | C | . | . | . | . | . | C | T | T | G | C | . | A | . | A | T | T | A | C | . | G | A | C | C | G | . | G | . | . | G | C | . | C | . | . | . | . | . | . |
| AB086102 | *Gallus gallus* | . | . | . | C | T | . | . | . | T | C | . | . | . | . | . | . | . | . | . | C | . | . | . | . | C | . | . | . | . | . | C | . | T | G | . | C | . | . | . | . | . | C | . | . | . | . | . | . |

## Designing the blocking oligonucleotide

To block amplification of DNA of Eurasian otters by a universal PCR assay targeting the 12S rRNA gene of vertebrates [17], we designed a Eurasian otter-specific blocking oligonucleotide according to the method reported elsewhere [16]. The designed oligonucleotide OBS1 with a 3-carbon spacer at the 3'-end specifically binds to and blocks amplification of the12S rRNA gene region of the Eurasian otter (Table 1). It was designed not to block amplification of 12S rRNA gene of its potential preys (Table 1).

To confirm the blocking efficacy, PCR assays with or without OBS1 targeting the 12S rRNA gene region of vertebrates with primers 12SV5F and 12SV5R [17] were performed using synthesized *Lutra lutra*'s DNA as a template. The DNA template containing priming sites of 12SV5F, 12SV5R, and OBS1 was synthesized from 499th to 634th nucleotide positions of the mitochondrial genome of *Lutra lutra* (FJ236015.1) and cloned into the pTOP Blunt V2 vector (Enzynomics, Seoul, Korea) at Macrogen Incorporation (Seoul, Korea). Each PCR reaction mixture (30 μl) contained 0.26 nM of the synthesized DNA template of *Lutra lutra* and 0.08 μM of each primer in Premix TaqTM (Takara Bio Inc., Shiga, Japan) with or without 0.8 μM of the blocking oligonucleotide OBS1. The thermal condition was at 95˚C for 15 min for initial denaturation, followed by 55 cycles of 30 s at 95˚C for denaturation and 30 s at 50˚C for annealing. The PCR products were electrophoresed and visualized on a 2.0% (w/w) agarose gel (OmniPur Agarose, Merck, NJ, USA) prepared in 1×TAE buffer by staining with SYBR® Green I Nucleic Acid Gel Stain (Invitrogen, MA, USA).

## DNA sequencing

PCR was performed with three sets of taxon-specific primers (Table 2), which are 12SV5F and 12SV5R targeting the 12S rRNA gene region of vertebrates [17], 16SMAV-F and 16SMAV-R targeting the 16S rRNA gene region of invertebrates [15], and VF2_t1, FishF2_t1, FishR2_t1 and FR1d_t1 targeting the cytochrome *c* oxidase 1 (COI) gene for fishes [18]. The expected sizes of amplicons, excluding the lengths of primers, are 98 bp for 12S rRNA gene [17], 36 bp for 16S rRNA gene [15], and 631 bp for COI [18]. These primers were attached with the adapter sequences for Illumina MiSeq (Illumina, Inc., CA, USA). Each PCR was performed in a 30 μl mixture containing 2 μl of DNA extract in Premix TaqTM (Takara Bio Inc., Shiga, Japan). For the vertebrate-specific 12S rRNA gene sequencing assay, 0.08 μM of each primer and 0.8 μM of each of blocking oligonucleotides OBS1 and HomoB [15] were added. The

**Table 2. Primers and blocking nucleotides used in this study.**

| Taxon | Target | Primer name | Type | Primer sequence 5'–3' | Ref. |
|---|---|---|---|---|---|
| Vertebrates | 12S rRNA gene | 12SV5F | Forward | TAGAACAGGCTCCTCTAG | [17] |
| | | 12SV5R | Reverse | TTAGATACCCCACTATGC | [17] |
| | | OBS1 | Eurasian otter blocking | CTATGCTCAGCCCTAAACATA GATAGCTTACATAACAAAAC TATCTGC–C3 | This study |
| | | HomoB | Human blocking | CTATGCTTAGCCCTAAACCTCAACAGTTAAATCAACAAAACTGCT–C3 | [15] |
| Invertebrates | 16S rRNA gene | 16SMAV-F | Forward | CCAACATCGAGGTCRYAA | [15] |
| | | 16SMAV-R | Reverse | ARTTACYNTAGGGATAACAG | [15] |
| | | MamMAVB1 | Mammal blocking | CCTAGGGATAACAGCGCAATCCTATT–C3 | [15] |
| Fishes | COI gene | VF2_t1 | Forward | TGTAAAACGACGGCCAGTCAACCAACCACAAAGACATTGGCAC | [18] |
| | | FishF2_t1 | Forward | TGTAAAACGACGGCCAGTCGACTAATCATAAAGATATCGGCAC | [18] |
| | | FishR2_t1 | Reverse | CAGGAAACAGCTATGACACTTCAGGGTGACCGAAGAATCAGAA | [18] |
| | | FR1d_t1 | Reverse | CAGGAAACAGCTATGACACCTCAGGGTGTCCGAARAAYCARAA | [18] |

additional assay was conducted without OBS1 and HomoB to check for unintended inhibition of amplification of prey DNA by these oligonucleotides. The thermal condition was at 95˚C for 15 min for initial denaturation, followed by 55 cycles of 30 s at 95˚C for denaturation and 30 s at 50˚C for annealing. For the invertebrate-specific assay, 0.2 µM of each primer and 2 µM of a blocking oligonucleotide for mammals (MamMAVB1) [15] were added. The thermal condition was at 95˚C for 15 min for initial denaturation, followed by 55 cycles of 30 s at 95˚C for denaturation and 30 s at 55˚C for annealing. For the COI assay, 0.167 µM of each primer were added. The thermal condition was at 95˚C for 15 min for initial denaturation, followed by 35 cycles of 30 s at 95˚C, 40 s at 52˚C and 1 min at 72˚C, followed by a final extension step at 72˚C for 10 min. The amplicons were purified and indexed using the Nextera XT Index kit (Illumina). Each of the purified indexed amplicons was normalized, pooled, thermally-denatured, and loaded onto a v3 600 cycle-kit reagent cartridge (Illumina) for 2 × 300 bp paired-end sequencing by Illumina MiSeq. Raw sequence data are available at NCBI under the Bio-Project accession numbers PRJNA497180, PRJNA548047, PRJNA548048, and PRJNA565647.

## DNA sequence processing and analyses

The raw sequence reads were quality trimmed, and the adapter sequences were removed using Trimmomatic v0.33 [21]. The remaining high-quality sequence reads of 12S and 16S rRNA genes were processed in OBITools [22]. The *illuminapairedend* command was used to assemble the paired-end reads followed by the *obigrep* command to remove unaligned sequences. As the barcodes had been already removed and sequences were already demultiplexed, we run the *ngsfilter* command with -:- option in place of the barcode tag sequence. The sample information was added as attributes in sequence headers of each sample, and the resultant sequences were concatenated and dereplicated using the *obiuniq* command. The reads shorter than 80 bp for 12S rRNA gene and 20 bp for 16S rRNA gene were removed using the *obigrep* command. The *obiclean* command was used to remove the erroneous reads. The remaining sequences were taxonomically assigned using the *ecotag* command against custom-made 12S and 16S rRNA gene sequences reference databases. These databases were built by extracting relevant regions of 12S and 16S rRNA gene from EMBL nucleotide database (release 138, February 21, 2019) using the *ecoPCR* program [23]. The *ecotag* command assigned the query sequence to the least common ancestor in case identification was ambiguous among sibling taxa. For the COI gene sequence analyses, the forward and reverse reads were not joined since the amplicon length (631 bp) was too long to be joined by 2×300 bp paired-end sequencing of Illumina Miseq. They were analyzed separately in mothur v1.40.5 [24]. First, identical reads were binned to generate a set of unique sequences, each with an identical length and nucleotide sequence. Next, the chimeric reads were removed by VSEARCH [25]. Finally, the remaining sequences were classified using the method reported by Wang et al. [26] in *classify.seq* command with a kmer search (kmer length of 8) against MIDORI database for COI gene [27].

## Statistical analyses

R version 3.6.0 was used for statistical analyses. The vegan package version 2.5–6 [28] was used to check for possible changes in the sequence structures by the blocking oligonucleotides OBS1 and for the difference in fish memberships detected by sequencing of two different loci of 12S rRNA and COI genes. For the first purpose, the Bray-Curtis dissimilarities of unique sequence structures were characterized and compared across the 12S rRNA gene libraries prepared with vs. without blocking oligonucleotides OBS1 and HomoB after excluding the sequence reads assigned to *Lutra lutra* and *Homo sapiens*. For the second purpose, the Jaccard distances were calculated to characterize the differences of memberships of fish genera

detected by 12S rRNA vs. COI gene sequencing analyses. Principal coordinate analysis plot and cluster dendrogram were created based on the Bray-Curtis and Jaccard distances calculated for the first and second purposes, respectively. Permutational multivariate analysis of variance (PERMANOVA) tests were conducted and followed by Kruskal-Wallis rank sum tests with *post hoc* Wilcoxon rank-sum tests with Bonferroni correction to compare pair-wise intra- and inter-sample distances to assess the degrees of variabilities across the samples and across the library preparation methods. *P*-values less than 0.05 were considered statistically significant.

## Results

### Sequencing statistics

The sample O4 was not amplified for 16S rRNA gene sequencing, resulting in n = 6 for the invertebrate analysis. After excluding the reads shorter than 80 bp, 727,245 and 417,605 joined paired-end reads were obtained for the vertebrate 12S rRNA gene sequencing with and without blocking oligonucleotides OBS1 and HomoB, respectively (S2 Table and S3 Table). After excluding the reads shorter than 20 bp, a total of 355,759 sequence reads were obtained by the invertebrate 16S rRNA gene sequencing (S4 Table). In total, 53,468 forward and 53,423 reverse reads of COI gene sequences were obtained (S5 Table). The rarefaction curves appear to reach asymptotes for most libraries, except for those with the small number of sequence reads (e.g., O1 for 16S rRNA gene) (S3 Fig).

### Performance of blocking oligonucleotide OBS1

The blocking oligonucleotide OBS1 statistically significantly inhibited the amplification of the predator DNA ($p<0.05$; Wilcoxon signed rank test), as we found only up to 0.21% of the 12S rRNA gene sequence reads assigned to *Lutra lutra* by the assay with OBS1 while 10–80% of the reads assigned without OBS1 (Table 3). We confirmed the absence of amplification of the otter's DNA by the conventional vertebrate-specific PCR assay with OBS1 (Fig 2). Additionally, we confirmed that the variabilities in unique sequence structures were smaller across the library preparation methods (i.e., with vs. without OBS1 and HomoB) than across the samples (Fig 3), indicating little inhibitory effects on prey DNA by OBS1 and HomoB. We found few human DNA sequences in our dataset (Table 3).

### Vertebrate 12S rRNA gene

The vertebrate families identified in spraints are shown in Fig 4A and 4B. Based on the assay with OBS1 and HomoB, the most abundant sequences were from the family of Cyprinidae

**Table 3. Number of 12S rRNA gene sequence reads assigned to *Lutra lutra* and *Homo sapiens*.** The results for the libraries prepared with or without blocking oligonucleotides OBS1 and HomoB are shown.

| Sample ID | Without OBS1 and HomoB | | | With OBS1 and HomoB | | |
|---|---|---|---|---|---|---|
| | Total | *Lutra lutra* | *Homo sapiens* | Total | *Lutra lutra* | *Homo sapiens* |
| O1 | 35183 | 16602 (47%) | 1 (0.003%) | 70154 | 149 (0.212%) | 0 (0%) |
| O2 | 70253 | 55973 (80%) | 5 (0.007%) | 98791 | 62 (0.063%) | 1 (0.001%) |
| O3 | 69078 | 22845 (33%) | 6 (0.009%) | 97201 | 55 (0.057%) | 0 (0%) |
| O4 | 98825 | 57136 (58%) | 165 (0.167%) | 137217 | 0 (0%) | 0 (0%) |
| O5 | 60031 | 5952 (10%) | 7 (0.012%) | 122219 | 0 (0%) | 0 (0%) |
| O6 | 26984 | 5644 (21%) | 7 (0.026%) | 101590 | 55 (0.054%) | 0 (0%) |
| O7 | 57251 | 5823 (10%) | 0 (0%) | 100073 | 0 (0%) | 0 (0%) |

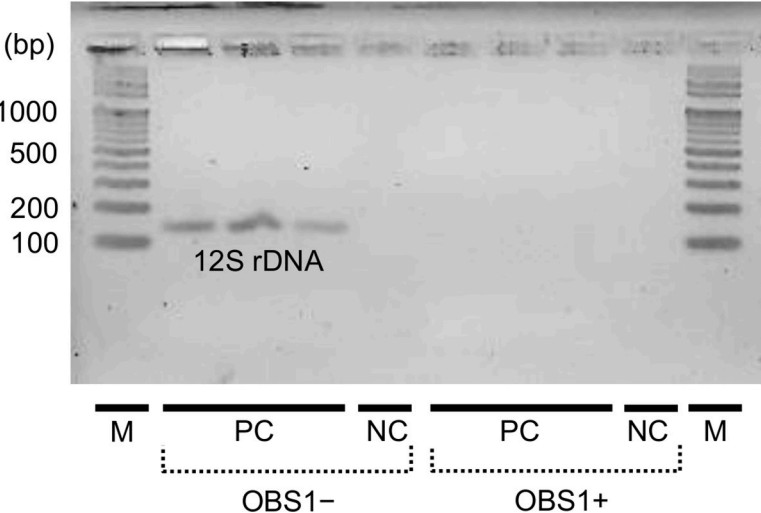

**Fig 2. The vertebrate-specific PCR assays with or without the blocking oligonucleotide OBS1.** The assays were performed to amplify the synthesized otter's DNA. Abbreviations: M, DNA marker; OBS1+, with OBS1; OBS1−, without OBS1; PC, positive control with the otter's DNA; and NC, negative control without the otter's DNA.

(55%), followed by Channidae (17%), Centrarchidae (9.5%), Mugilidae (7.6%), Ranidae (3.3%), Gobiidae (2.7%), Anguillidae (1.8%), Acheilognathidae (1.0%), and Osphronemidae (0.8%). Except the family Ranidae (amphibian), all other families belong to fishes. At the

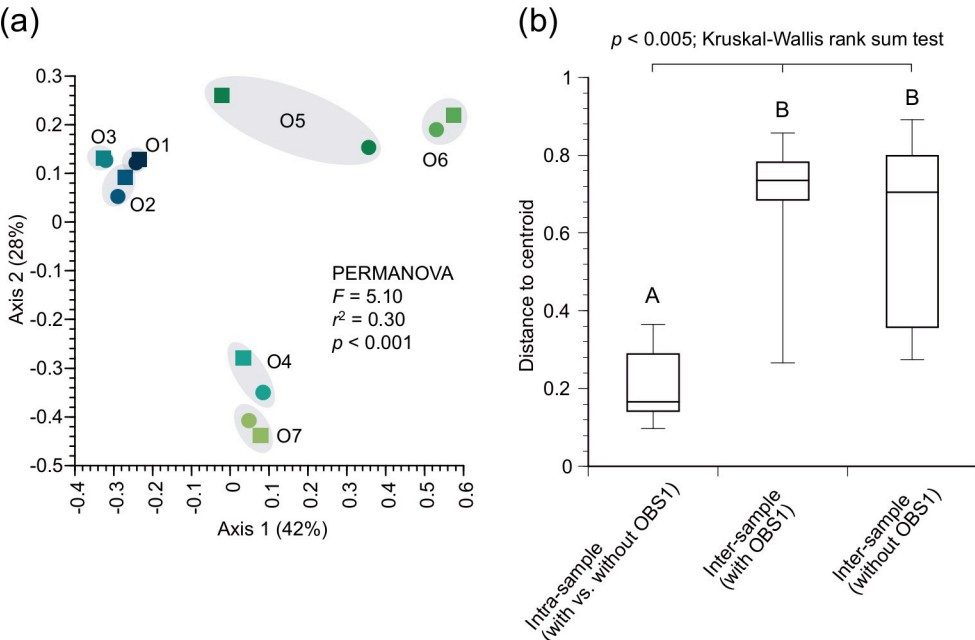

**Fig 3. Similarity of 12S rRNA gene sequence structures across libraries.** The distances are based on the Bray-Curtis dissimilarities of the unique sequence structures. The reads from *Lutra lutra* and *Homo sapiens* are excluded from calculation of the dissimilarity distances. (a) Principal coordinate analysis plot. Circle and squares symbols represent libraries prepared with and without blocking oligonucleotides OBS1 and HomoB, respectively. PERMANOVA test was performed by grouping libraries from the same samples. (b) Boxplot showing the distributions of pair-wise distances across libraries. The intra-sample distances obtained with and without OBS1 and HomoB, the inter-sample distances obtained with OBS1 and HomoB, and the inter-sample distances obtained without OBS1 and HomoB are shown. The results of *post hoc* pair-wise comparisons based on Wilcoxon rank-sum tests with Bonferroni correction are shown with significant differences indicated by different letters.

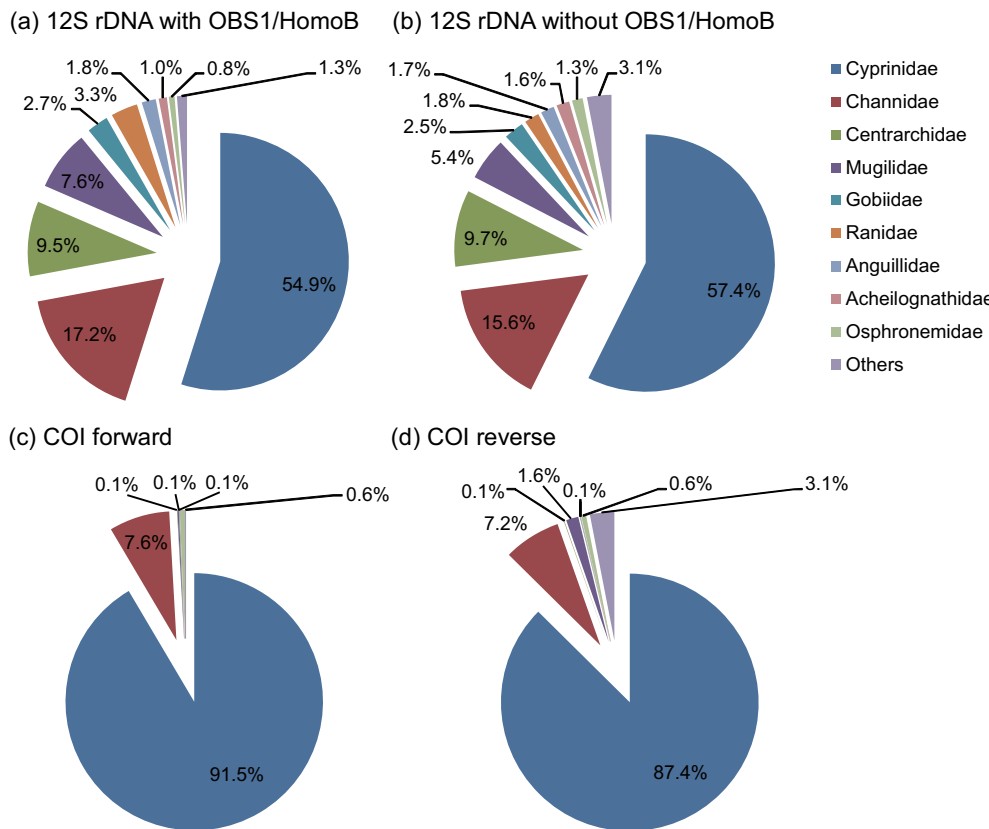

**Fig 4. Mean relative abundances of vertebrate and fish families detected from sprants (n = 7).** (a) Vertebrate families detected by 12S rRNA gene sequencing with OBS1 and HomoB. (b) Vertebrate families detected by 12S rRNA gene sequencing without OBS1 and HomoB. The relative abundances were calculated by excluding the reads assigned to the family Mustelidae to which *Lutra lutra* belongs. (c) Fish families detected from the forward reads of COI gene sequencing. (d) Fish families detected from the reverse reads of COI gene sequencing. For the COI data, only the reads assigned to the clade Actinopteri are included to calculate relative abundances.

species or genus levels, 11 taxa were identified with more than 1% of relative abundances from at least one of the analyzed libraries (Fig 5A and 5B). Based on the assay with OBS1 and HomoB, the detected taxa and their mean relative abundances were *Carassius auratus* (crucian carp) (47%), *Channa argus* (northern snakehead) (17%), *Lepomis macrochirus* (bluegill) (8.5%), *Mugil cephalus* (flathead grey mullet) (7.2%), *Anguilla japonica* (glass eel) (1.8%), *Cyprinus capio* (common carp) (1.5%), *Acheilognathus barbatus* (bitterling fish) (1.0%), *Macropodus* spp. (paradise fishes) (0.8%), and *Micropterus salmoides* (largemouth bass) (0.8%). The non-fish taxa detected were *Pelophylax nigromaculatus* (water frog) (2.7%) and *Rana* spp. (brown frogs) (0.5%). The vertebrate taxa identified at the species level and their sibling species included in the reference database are listed in S6 Table.

## Invertebrate 16S rRNA gene

The invertebrate families detected from sprants are shown in Fig 6. Varunidae was the most abundant (46%), followed by Palaemonidae (18%), Reduviidae (11%), Chironomidae (3.4%), Plumatellidae (2.6%), Atyidae (1.2%), Cicadellidae (1.2%), and Gyrinidae (1.0%). At the species or genus levels, *Helicana* spp. (mud flat crabs) was the most abundant (46%) and detected from 5 out of 6 sprants (Fig 7). Other species detected included shrimps such as *Palaemon*

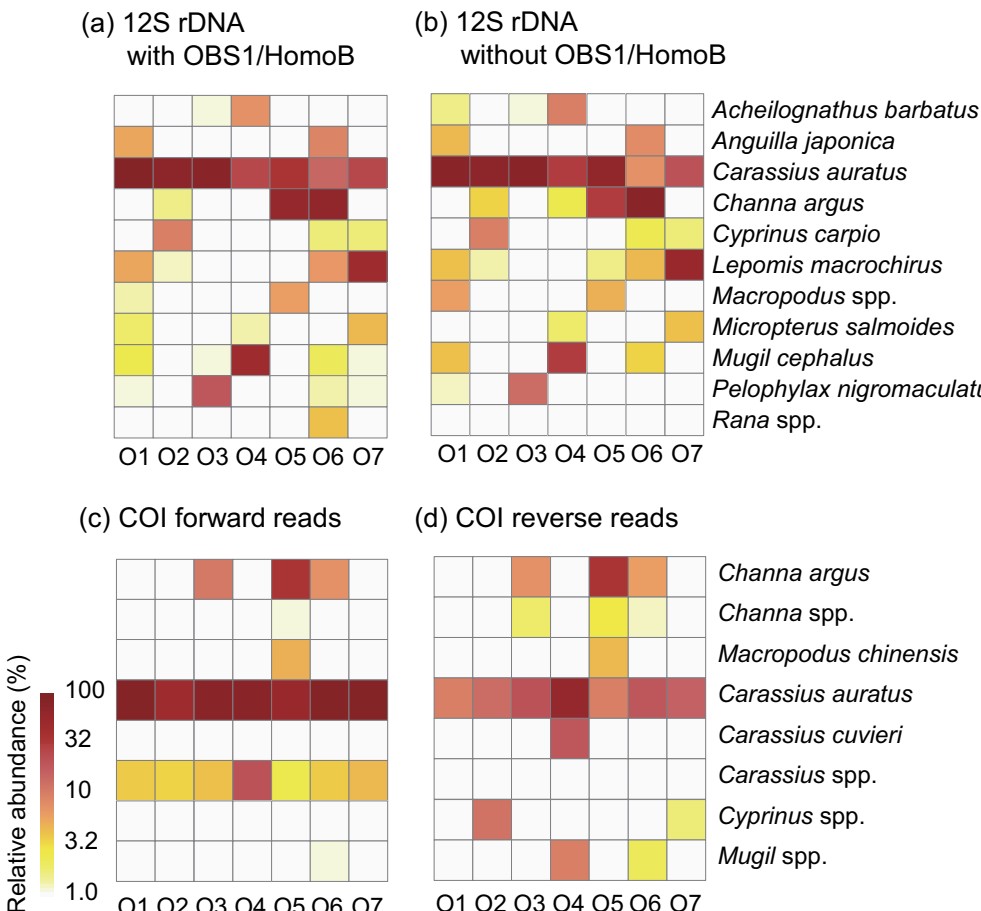

**Fig 5. Relative abundances of species or genera of vertebrates and fishes detected from spraints.** Taxa detected with more than 1% of relative abundances from at least one of the analyzed libraries are shown. (a) Vertebrate taxa detected by 12S rRNA gene sequencing with OBS1 and HomoB. (b) Vertebrate taxa detected by 12S rRNA gene sequencing without OBS1 and HomoB. The relative abundances were calculated by excluding the reads assigned to the family Mustelidae to which *Lutra lutra* belongs. (c) Fish taxa detected from the forward reads of COI gene sequencing. (d) Fish taxa detected from the reverse reads of COI gene sequencing. For the COI data, only the reads assigned to the clade Actinopteri are included to calculate relative abundances.

*paucidens* (14%) and *Palaemon modestus* (3.9%), *Paratanytarsus grimmii* (chironomid) (3.4%), *Macrogyrus oblongus* (whirligig beetle) (1.0%), and the ostracod species such as *Physocypria* sp. X IK-2017 (0.7%) and *Cypridopsis vidua* (0.2%). The invertebrate taxa identified at the species level and their sibling species included in the reference database are listed in S7 Table.

## Fish COI gene

The large numbers of non-fish sequence reads were detected, i.e., 45,613 forward reads (85%) and 44,817 reverse reads (84%) (S5 Table), due to lack of selectivity of the primers for fish detection from environmental samples [18]. Nonetheless, the remaining 7,855 forward reads (15%) and 8,606 reverse reads (16%) were assigned to Actinopteri, a clade encompassing most species of fishes. The identified fish families were Cyprinidae, Channidae, Osphronemidae, Mugilidae, Centrarchidae, and Gobiidae (Fig 4C and 4D), which included *Carassius* spp. including *Carassius auratus* and *Carassius cuvieri* (white crucian carp), *Channa* spp. including

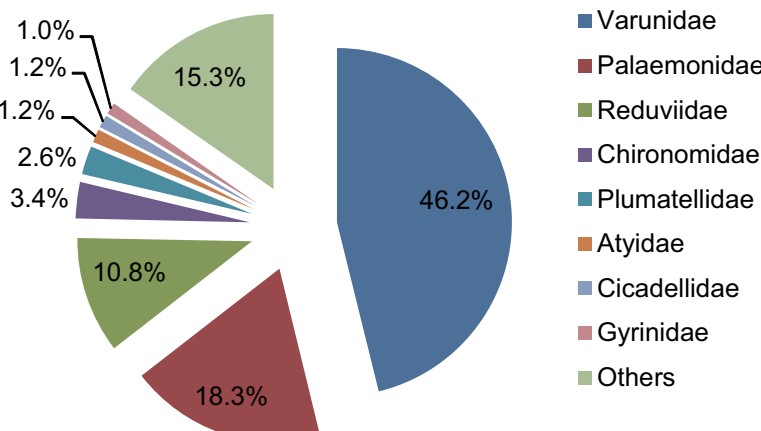

**Fig 6. Mean relative abundances of invertebrate families detected from spraints (n = 6).** Invertebrate families detected by 16S rRNA gene sequencing are shown.

*Channa argus*, *Cyprinus* spp., *Macropodus chinensis*, and *Mugil* spp. (Fig 5C and 5D). The fish taxa identified at the species level and their sibling species included in the database are listed in S8 Table.

The fish genera detected by the COI sequencing appear to be similar to those by the 12S rRNA gene sequencing (Fig 8), with examples of the identical genera detected in the samples O2 and O5 between the two different sequencing loci (S9 Table). Fig 8 shows that the variabilities in the memberships of detected fish genera were smaller across the sequencing loci (i.e., 12S rRNA gene vs. COI) than across the samples, with a statistical significance observed between the intra-sample distances based on the two different loci and the inter-sample distances based on 12S rRNA gene sequencing, indicating that the detected fish genera are in agreement between the two locus analyses in terms of detected fish memberships.

## Discussion

In this study, we designed a Eurasian otter-specific blocking oligonucleotide (OBS1) for their diet analyses based on 12S rRNA gene sequencing for vertebrates. The use of blocking

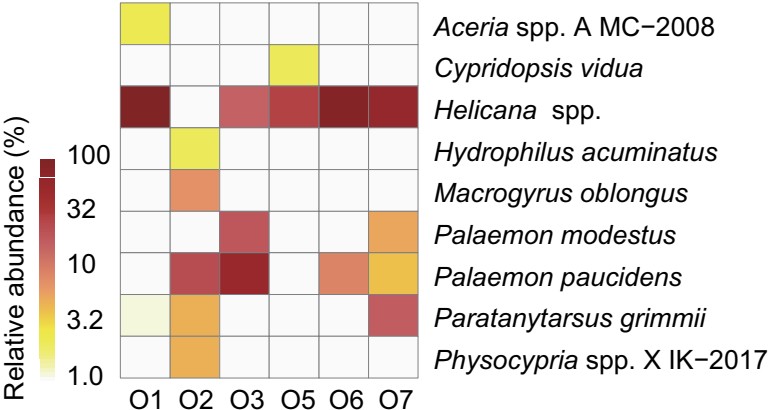

**Fig 7. Relative abundances of species or genera of invertebrates detected from spraints.** Taxa detected with more than 1% of relative abundances from at least one of the analyzed libraries are shown. Invertebrate taxa detected by 16S rRNA gene sequencing are shown.

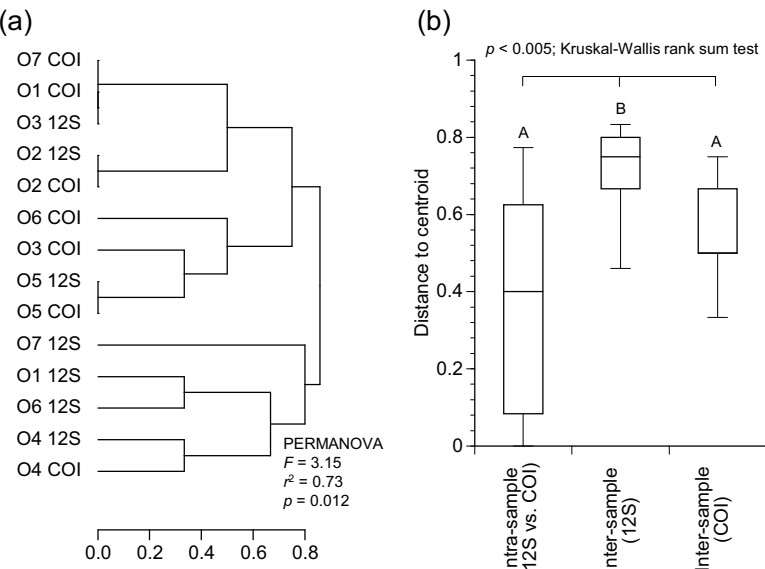

**Fig 8. Similarity of fish memberships detected by 12S rRNA and COI gene sequencing.** The Jaccard distances are shown. The fish genera detected with more than 1% of relative abundance from any of the analyzed libraries (S9 Table) are included for the analysis. (a) Cluster dendrogram with the result of PERMANOVA test by grouping libraries from the same samples but analyzed by the different target loci (i.e., 12S rRNA vs. COI genes). (b) Boxplot showing the distributions of pair-wise distances across libraries. The intra-sample distances obtained by 12S rRNA vs. COI gene sequencing, the inter-sample distances obtained by 12S rRNA gene sequencing, and the inter-sample distances obtained by COI gene sequencing are shown. The results of *post hoc* pair-wise comparisons based on Wilcoxon rank-sum tests with Bonferroni correction are shown with significant differences indicated by different letters.

oligonucleotides is an effective solution to suppress amplification of DNA fragments derived from the predators such as brown bears [15], leopard cats [11], penguins [29], and seals [30]. It has been shown that blocking oligonucleotides can increase sensitivity of detecting rare preys by suppressing amplification of predator DNA [11]. The blocking oligonucleotide OBS1 designed to specifically bind to the 12S rRNA gene region of the Eurasian otter (Table 1) successfully inhibited amplification of the synthesized DNA fragments of the Eurasian otter (Fig 2), with only up to 0.21% of the total reads assigned to the otter by the vertebrate-specific 12S rRNA gene sequencing (Table 3). We confirmed that the types of fishes detected by the 12S rRNA gene sequencing were congruent with those detected by the COI gene sequencing in terms of their memberships (Fig 8 and S9 Table). Moreover, we confirmed that the variabilities in unique sequence structures were smaller across the library preparation methods (i.e., with vs. without OBS1) than across the samples (Fig 3), suggesting little inhibitory effects on prey DNA by the designed blocking oligonucleotide OBS1.

Our analyses of 12S rRNA gene sequences of vertebrates showed that fishes (95%) were the primary diet, followed by amphibians (3.3%) for the Eurasian otters inhabiting a marshy estuary area of South Korea in the early summer (Fig 4A). Previous studies [31–33] demonstrated that 12S rRNA gene sequences provide accurate identification of most of fishes including teleosts at species level, highlighting the 12S rRNA gene as an ideal marker for diet analyses of Eurasian otters. Additionally, we confirmed that fish species identified in this study are in agreements with fish species detected from sprints by a previous study [10] and spatial distributions of their habitats in Korea [34].

Our observation of fishes as dominant preys of the Eurasian otters in South Korea is in congruence with previous studies showing dominance of fish preys of the Eurasian otters in Europe based on conventional morphological identification methods [3, 4]. Similar to the

previous findings [35–37], the majority of the detected fishes belonged to the family Cyprinidae (Fig 4) among which *Carassius auratus* (crucian carp) was the most dominant species detected across all spraints (Fig 5). A previous study also reported that Cyprinidae was the most dominant diet by the Eurasian otters along a river basin in South Korea [10]. *C. auratus* dominates fresh water ecosystems in South Korea [38], and therefore these might represent an abundant prey in their local habitats. The other freshwater species detected in this study included *Channa argus* (northern snakehead), *Lepomis macrochirus* (bluegill) and *Cyprinus capio* (common carp). These freshwater species are known to inhabit Korea [39].

Euryhaline species, such as *Mugil cephalus* (flathead grey mullet), were also detected, and they are known to inhabit freshwater and seawater ecosystems in Korea [40]. The detection of these species seems reasonable since the spraints were collected in an estuary area with a freshwater stream flowing into a regulating brackish lake neighboring the Yellow Sea (Fig 1). *M. cephalus* possibly migrates into the freshwater stream in the area where the Eurasian otters inhabit, and/or the Eurasian otters possibly feed in the areas of the brackish lake where *M. cephalus* inhabits.

Introduced species such as *Lepomis macrochirus* (bluegill) and *Micropterus salmoides* (largemouth bass) were detected (Fig 5A and 5B), with their respective mean relative abundances of 8.5% and 0.8%. *L. macrochirus* is native to North America, and was first imported to Japan in 1960 [41]. Later, a group of *L. macrochirus* was imported from Japan and released into a reservoir in Korea from which it was dispersed and became a dominant species in freshwater systems in Korea [41]. The Eurasian otters fed on these exotic species, but their quantities, i.e., 8.5% for *L. macrochirus* and 0.8% for *M. salmoides*, were smaller than those of the native species such as *Carassius auratus* (crucian carp) (47%) and *Channa argus* (northern snakehead) (17%), which was similar to the previous finding that the exotic fishes, including *Lepomis gibbosus* (pumpkinseed), were preyed at disproportionately lower levels than the native fishes, including *Anguilla anguilla* (European eel), by the Eurasian otters in England [42].

Non-fish vertebrates were also detected, which included frogs such as *Pelophylax nigromaculatus* (water frog) and *Rana* spp. (brown frogs) (Fig 5A and 5B). Eurasian otters prey on amphibians when fishes become scarce [37]. Both of these frog species inhabit Korea [43], and a study reported *Rana* spp. in spraints collected along a river basin in Korea [10]. However, they were not abundant, with mean relative abundances of 2.7% for *Pelophylax nigromaculatus* and 0.5% for *Rana* spp., suggesting that frogs are not primary diets of the Eurasian otters during this time of the year at our sampling site possibly because of the sufficient availability of fishes.

Among invertebrates, *Helicana* spp. (mud flat crabs) were the most abundant (46%) (Fig 7), followed by shrimps such as *Palaemon paucidens* and *Palaemon modestus*. *Helicana* spp. are widely distributed in coastal wetlands in Korea [44, 45]. Similarly, *P. modestus* and *P. paucidens* are endemic in freshwater ecosystems of Korea [46]. Eurasian otters are known to feed on crabs and other crustaceans [37, 47, 48]. Additionally, the insects were detected from some spraints with relatively high abundances. Insects are known to be fed by Eurasian otters [3, 37]. We detected *Macrogyrus oblongus* (whirligig beetle) with 6.2% of relative abundance from the sample O2. This water beetle is known to inhabit parts of Oceania and South America, but not East Asia [49]. This suggests potential misidentification of a closely related genus such as *Dineutus* that is known to inhabit Korea [49]. The family Reduviidae (assassin bugs) was also detected (Fig 6), with a high relative abundance (65%) in the sample O5, but the sequences were not identified down to the genus level. Reduviidae is unlikely a common prey for the Eurasian otters since no such observation has been reported by previous research. The otter might opportunistically prey Reduviidae that might be present in its local habitat.

There are several limitations in our DNA metabarcoding-based diet analyses. First, caution must be taken when drawing conclusions of relative biomass consumed based on relative sequence abundance in environmental DNA since the relationship between sequence abundance and biomass can often vary and be weakly correlated due to differences in amplification efficiency of primers particularly in COI gene [50–52]. A semiquantitative estimate of diet composition could be obtained in future studies after applying the robust filtering procedure described by Corse et al. [53]. It is also impossible to compare relative importance between invertebrates and vertebrates based on our multiple locus approach. Second, we detected milli-meter-size organisms such as *Paratanytarsus grimmii* (chironomid) (3.4%), and the species of ostracods *Physocypria* sp. (0.7%) and *Cypridopsis vidua* (0.2%) (Fig 7). Due to their small sizes, these organisms were unlikely predated by the Eurasian otters. Instead, these organisms were likely incidentally ingested by and/or attached to larger organisms predated by the otters. The secondary predation is also possible, for instance, that the Eurasian otters feed on piscivorous fishes (e.g., northern snakehead) that previously fed on smaller fishes (e.g., crucial carp) and contained them in their guts. We do not know such relationships. Other potential limitations include the inability of detecting possible cannibalism [54] because, unless special approaches, such as the so-called the variant-centered approach for haplotype detection [53, 55], are employed, DNA metabarcoding cannot distinguish DNA fragments from cannibalized and cannibalizing individuals. Finally, a limitation of this study was small sample size as the accumulation curves of the numbers of detected prey genera do not appear to reach to the asymptotes (S4 Fig). Future research should include more samples to capture the entire picture of the diet profiles in this study site.

## Conclusion

In this study, we applied HTS-based DNA metabarcoding analyses targeting vertebrate 12S rRNA gene, invertebrate 16S rRNA gene, and fish COI gene sequences to characterize diet profiles of the Eurasian otters inhabiting a marshy estuary area of South Korea. In particular, we developed a Eurasian otter-specific blocking oligonucleotide (OBS1) for 12S rRNA gene sequencing for vertebrates. Overall, our study demonstrated that HTS-based DNA metabarcoding analyses were useful to provide in-depth information regarding their diet compositions. Our multi locus approach showed that the combination of 12S and 16S rRNA gene sequencing could cover a broad range of invertebrate and vertebrate prey animals of the Eurasian otters, including fishes detected by the COI gene sequencing. Our results suggest that the 12S and 16S rRNA gene sequencing analyses suffice for diet analyses of the Eurasian otters, with optional COI gene sequencing for fish-specific detection and identification. In this study, spraint samples were collected at one location in one season, and the diet profiles might differ spatiotemporally [3, 4]. By using HTS-based metabarcoding analyses, future research will explore detailed taxonomies of the otters' diets across locations and seasons.

## Supporting information

**S1 Fig. Spraints collected in this study.**
(TIF)

**S2 Fig. DNA bands of the PCR amplicons by the Eurasian otter-specific primers LutcytF and LutcytR.**
(TIF)

**S3 Fig. Rarefaction curves for observed unique sequences.** (a) 12S rRNA gene for verte-brates. The letter "b" in sample IDs indicates libraries prepared with blocking oligonucleotides

OBS1 and HomoB. The sample IDs without the letter "b" indicate the results prepared without OBS1 and HomoB. (b) 16S rRNA gene for invertebrates. (c) Forward reads of COI gene sequences. (d) Reverse reads of COI gene sequences. For the COI data, the reads of non-fish sequences are also included.
(TIF)

**S4 Fig. Cumulative distributions of the numbers of detected genera.** The genera detected with more than 1% of relative abundance are included.
(TIF)

**S1 Table. DNA concentrations of extracts from spraint samples.**
(XLSX)

**S2 Table. Number of high-quality vertebrate 12S rRNA gene sequence reads prepared with blocking oligonucleotides OBS1 and HomoB.** The reads shorter than 80 bp are excluded.
(XLSX)

**S3 Table. Number of high-quality vertebrate 12S rRNA gene sequence reads prepared without blocking oligonucleotides OBS1 and HomoB.** The reads shorter than 80 bp are excluded.
(XLSX)

**S4 Table. Number of high-quality invertebrate 16S rRNA gene sequence reads.** The reads shorter than 20 bp are excluded.
(XLSX)

**S5 Table. Number of high-quality COI gene sequence reads.** The numbers of total COI gene sequence reads are listed with those of the sequences assigned to the clade Actinopteri.
(XLSX)

**S6 Table. List of species in the reference database of 12S rRNA gene sequences.** The species identified in this study and their sibling species included in the database are listed.
(XLSX)

**S7 Table. List of species in the reference database of 16S rRNA gene sequences.** The species identified in this study and their sibling species included in the database are listed.
(XLSX)

**S8 Table. List of species in the reference database of COI sequences.** The species identified in this study and their sibling species included in the database are listed.
(XLSX)

**S9 Table. Comparison of fish genera detected by 12S rRNA gene and COI sequencing.** The genera detected with more than 1% of relative abundance of any of the analyzed libraries are listed. The numbers "1" represents detected while "0" represents not detected. For the COI data, a taxon was assumed to be detected if it was detected with more than 1% of relative abundance from the forward and/or reverse sequence reads.
(XLSX)

## Author Contributions

**Conceptualization:** Kyung Yeon Eo, Woo-Shin Lee, Junpei Kimura, Naomichi Yamamoto.

**Data curation:** Priyanka Kumari, Ke Dong.

**Formal analysis:** Priyanka Kumari, Ke Dong.

**Funding acquisition:** Woo-Shin Lee, Junpei Kimura, Naomichi Yamamoto.

**Investigation:** Kyung Yeon Eo, Junpei Kimura, Naomichi Yamamoto.

**Project administration:** Naomichi Yamamoto.

**Validation:** Naomichi Yamamoto.

**Visualization:** Priyanka Kumari, Naomichi Yamamoto.

**Writing – original draft:** Priyanka Kumari, Naomichi Yamamoto.

**Writing – review & editing:** Ke Dong, Kyung Yeon Eo, Woo-Shin Lee, Junpei Kimura.

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
