## [Decision Letter · Decision Letter 0]

3 Sep 2019

PONE-D-19-19235

DNA metabarcoding-based diet survey for the Eurasian otters (Lutra lutra) in a marshy estuary area of South Korea

PLOS ONE

Dear Dr. Yamamoto,

Thank you for submitting your manuscript to PLOS ONE. After careful consideration, we feel that it has merit but does not fully meet PLOS ONE’s publication criteria as it currently stands. Therefore, we invite you to submit a revised version of the manuscript that addresses the points raised during the review process.

Thank you for submitting your research to PLoS ONE journal. We have now received two referee reports for your manuscript. They feel that your findings about DNA metabarcoding-based diet survey for the Eurasian otters are potentially interesting, but also raise criticisms concerning about the main topic of this manuscript based on small sampling size of the study. I agree with their concerns, and major re-formulation of the study aim will be required for this manuscript. Therefore, my editorial decision is major revision. And the referees provide excellent suggestions, which should all be addressed.

We would appreciate receiving your revised manuscript by Oct 18 2019 11:59PM. To enhance the reproducibility of your results, we recommend that if applicable you deposit your laboratory protocols in protocols.io, where a protocol can be assigned its own identifier (DOI) such that it can be cited independently in the future. For instructions see: http://journals.plos.org/plosone/s/submission-guidelines#loc-laboratory-protocols

We look forward to receiving your revised manuscript.

Kind regards,

Zuogang Peng, Ph.D.

Academic Editor

PLOS ONE

Journal Requirements:

1. We note that you are reporting an analysis of a microarray, next-generation sequencing, or deep sequencing data set. PLOS requires that authors comply with field-specific standards for preparation, recording, and deposition of data in repositories appropriate to their field. Please upload these data to a stable, public repository (such as ArrayExpress, Gene Expression Omnibus (GEO), DNA Data Bank of Japan (DDBJ), NCBI GenBank, NCBI Sequence Read Archive, or EMBL Nucleotide Sequence Database (ENA)). In your revised cover letter, please provide the relevant accession numbers that may be used to access these data. For a full list of recommended repositories, see http://journals.plos.org/plosone/s/data-availability#loc-omics or http://journals.plos.org/plosone/s/data-availability#loc-sequencing.

Reviewers' comments:

Reviewer's Responses to Questions

**Comments to the Author**

1. Is the manuscript technically sound, and do the data support the conclusions?

Reviewer #1: Partly

Reviewer #2: Partly

2. Has the statistical analysis been performed appropriately and rigorously? 

Reviewer #1: No

Reviewer #2: N/A

3. Have the authors made all data underlying the findings in their manuscript fully available?

Reviewer #1: Yes

Reviewer #2: Yes

4. Is the manuscript presented in an intelligible fashion and written in standard English?

Reviewer #1: Yes

Reviewer #2: Yes

5. Review Comments to the Author

Reviewer #1: This manuscript reports a study in which researchers used multiple universal metabarcoding primers to characterize the diets of Eurasian otter (Lutra lutra) and developed novel blocking primers to reduce the amount of sequencing effort lost on sequencing non-target predator DNA. The authors did a very good job succinctly outlining the importance and rationale for this research, and the methods are thoroughly and clearly outlined. However, the study in this manuscript is limited in scope, mainly serving as a proof of concept for the methodology presented, with the blocking primers for Lutra lutra being the main novel contribution this manuscript makes to the literature. The sample size is too small and the study was not designed to make any ecological comparisons, so I would recommend the authors rewrite parts of this paper to emphasize their testing of the methods and spend less time in the manuscript on trying to draw ecological conclusions from their limited data. Additionally, descriptive statistics that were reported may need to be recalculated based on rarified sequence data and I recommend more rigorous statistical testing to determine whether the sample size is adequate to draw any ecological conclusions from the data presented in this manuscript. Specific criticisms are listed below.

Title

I recommend reworking the title to focus more on the methodology used (mention the blocking primers)

Abstract

12S and 16S barcoding regions are typically classified as ribosomal RNA (rRNA), not mtDNA. This should be changed throughout the manuscript.

Introduction

Line 37 – Citation for the sentence about the limited resolving power of morphological methods?

Materials and methods

Line 65 – S1 Fig not included in supplementary materials

Lines 84-87 – This section is confusingly worded. If I understand correctly, there were three subsamples taken from each homogenized spraint, DNA was extracted from each subsample and then recombined for further analysis?

Lines 87-88 – Why was a different set of DNA used for the COI analysis?

Line 90 – S2 Fig not included in supplementary materials

Line 113 – What was the concentration/amount in ng of the template DNA used in the PCR reactions?

Line 144 – Were sequences rarefied by sample prior to analysis? If not, this could bias the calculations of relative abundance toward the observations in samples that simply sequenced better. Taking an equal-sized random subsample of sequences from each sample after erroneous reads have been filtered out would prevent this bias (e.g. subsample 68,053 reads from each 12S sample)

Results

Line 162 – More information could be provided on the predator sequences recovered (i.e. Were they detected in multiple samples?)

6-7 samples seems like a small sample size to claim to be a representative sample of the diets of this population. Accumulation curves of unique prey taxa observed with each sample would be helpful to visualize whether or not these samples are representative of the diet diversity of this population and could be used to help estimate what sample size would be necessary to get a representative sample in future studies.

Discussion

Lines 220-232 – Main results from all the analyses should be presented in the first paragraph(s). Success of the blocking primers should be one of the main takeaways

Lines 234-266 – The discussion is hampered by the overly detailed and speculative explanations of all of the prey taxa observed in the dataset. I recommend concentrating on the most common observations, any unexpected taxa, or taxa of relevant for conservation.

Lines 247-250 – Caution should be exercised when drawing conclusions of relative biomass consumed based on relative sequence abundance in complex DNA mixtures. Due to primer and sequencing bias, the relationship between biomass and sequence abundance is often weakly correlated and cannot be assumed without test mixtures of target DNA with known biomass contributions.

Lines 265-266 – This is somewhat speculative to assume this observation is a predation event. Could be contamination?

Lines 280-281 – Are there alternative explanations to predation in this case? Would members of this family utilize fecal resources for water, food, breeding, etc.?

Line 287 – Incidental ingestion is also a possible explanation

Lines 302-307 – A broader discussion of the advantages and limitations of using multiple universal primer sets would be useful.

Figures

Figure 1 – Main map needs a compass and scale bar. It is not clear to a non-Korean audience that the inset map is depicting South Korea without more geographical context. The extent indicator on the inset map does not line up with the extent of the main map. Labeling of water bodies on the main map would be helpful.

Reviewer #2: In this study authors aims to investigate diet of Eurasian otters, using metabarcoding of Luttra praints (n=7) from brackish environment in Korea. They targeted three different genes to detect the entire range diversity of prey (vertebrates and invertebrates). Authors also developed a blocking primer targeting predator DNA. Results showed that one teleost species (ie Carrassius auratus) is the main prey of the predator.

In this state, the ms may be criticizable by its sampling size. So, I think that authors should reformulate the aim of the study (“Investigate diet profiles of eurasian otters”) For me it is more a feasibility study/proof of concept, and analysis and discussion should be in that direction, and lesser to trophic ecology discussion; although I agree with authors, there is an important biological result: Lutttra seems to fed mostly on one species: Carassius auratus .

Here are some example of problematics that should be discuss in this sense:

1-What do you advice for future studies? Is the use of multi locus approach is relevant? Because there is a trade-off between the number of amplicons (that increase the financial cost) and the complementarity between them. In your case the three locus bring different information because they were designed to target different taxa, and your result is congruent with your expectations. It should be highlighted in the discussion. Also, is the 16S amplion that targeted the invertebrates is relevant for future studies of Luttra diet?

Also, the challenge when using different locus in metabarcoding in my sense is related to the comparison of results. Are results of 12S and COI dataset are similar, both in term of taxa identification (qualitative aspect) and relative abundance (quantitative aspect)? Incongruent? It is relevant to keep both of them for future studies? Also, is it possible with your protocol to compare the relative importance between invertebrates and vertebrates in the diet of Luttra?

2- Are the three genes enough informative to study the diet of Luttra? For example, as fishes are key prey for Luttra: Is the 12S and COI gene are sufficiently discriminant between sister species to allow identification of teleost at species level?

3- Are the reference sequence databank are sufficiently documented for the study area to allow a more important diet study of Luttra? Is a special barcoding effort of teleost (that seems to be the preferential prey of Luttra) species is necessary?

4-Is the use of blocking primer is efficiency and does not biased the result? See my remark below.

I have also some remark on Methodologies:

I did not understood why you did not merged forward and reverse sequencing for the COI dataset? Because of potential sequencing bias (eg. forward primer should sequenced better taxa and reverse primer sequenced better other taxa)? In general, forward/reverse primer sequencing is used when target amplicon is large. But in this ms, the two data set were not merged, and were not compared. You have to explain why you did not.

I have general remark and question on Material & Methods:

1- Why did you not use replicates sample?

2- Relative abundance of reads: if reads proportion is used for ribosomal genes in metabarcoding, it is more controversial when using COI gene. You should evoke this in your discussion and explain how you deal with your data in this context.

3- The use of blocking primer is useful, but difficult to design. In my experience, its specificity should always be check by in silico tests (as you did) but also by in vivo tests. For that, you should use positive control, like a mix of teleost and invertebrates DNA (target prey) and a negative control with only DNA targeted by the blocking primer (here the lutra lutra DNA), in order to ensure 1) that prey DNA is not blocked, and 2) to assess the efficiency of the blocking primer to inhibit the amplification of the predator DNA. Indeed, maybe you found few predator DNA just because the PCR primer did not work well on predator DNA. In this case, blocking primer is not essential and should be removed to avoid potential interaction with prey DNA. The validation of blocking primer need both approach.

See more on the interest to use positive and negative control in diet metabarcoding in Corse, E. et al. A from-benchtop-to-desktop workflow for validating HTS data and for taxonomic identification in diet metabarcoding studies. Mol Ecol Resour 17, e146–e159 (2017).

Minor remark

l.220 I think that you should moderate your conclusion. Indeed, are you sure that your PCR primer were not simply better for fish than for vertebrates? Is your blocking primer should inhibit some vertebrates DNA amplification?

l.255 Some terms should be check to avoid confusion: What do you mean by “fish susceptibility to predation” ? Prey selection? Also, what do you mean by relative abundance? Their relative abundance in the environment?

l.289/290 Not totally true. It’s possible to detect cannibalism if you use “a variant-centered metabarcoding data filtering”, rather than a “sequence group” filtration. When using COI, such filtering process allow to obtain the different haplotype of the same species; see in Corse et al., 2017. Using this approach, we detected intra species predation on trutta and Gobiidae, see Corse et al., 2019:

Corse, E. et al. One-locus-several-primers: A strategy to improve the taxonomic and haplotypic coverage in diet metabarcoding studies. Ecology and Evolution 9, 4603–4620 (2019).

l.291-293 This process named secondary predation.

l.296 I think you had to precise what type of blocking primer. Indeed, there is not only one type of blocking primer.

l.299-302 In my sense, you can not be sure with your data that it is your blocking primer that inhibit predator DNA amplification (see my general remark).

l.303 Your argument is good only if the COI dataset is representative of the fishes diversity in the diet of Luttra. So, are you sure that your COI primer allow to obtain a representative picture of the diversity of fishes in the diet? In fact, you have to discuss that your COI primer set allow to amplify all potentially fishes eaten by Luttra.

l.317 Please, check this: “…however [3, 4]”

Figures

Fig.1 Information are lacking. Provides some geographic landmarks such as cities, river and ocean names, area of study site, map scales.

Fig.2 This figure is hard to follow.

1) I think you have to choose, giving relative abundance in the text or in the figure. For me, relative abundance by taxa is redundant in a pie chart figure. Keep % only in the text.

2) You should remove color, or keep unique color code for the four pie chart.

3) You should indicate in a subtitle above each pie chart the data used. Example for a) 12S gene sequence b) 16S gene sequence ….

Fig.3 As for Fig 2 you should add subtitle. Also you can remove phylogenetical trees because they did not supply any crucial information and did not revealed true phylogenetical relationship.

6. PLOS authors have the option to publish the peer review history of their article (what does this mean?). If published, this will include your full peer review and any attached files.

Reviewer #1: No

Reviewer #2: Yes: Emmanuel Corse

---

## [Author Response · Author response to Decision Letter 0]

4 Oct 2019

The authors thank high-quality comments from the Reviewers #1 and #2 and constructive suggestion by the Editor. The comments were useful to improve our manuscript. Below, please find our responses to the reviewers’ comments. The page numbers in our responses refer to the clean version of our revised manuscript. 

Editor:

Thank you for submitting your research to PLoS ONE journal. We have now received two referee reports for your manuscript. They feel that your findings about DNA metabarcoding-based diet survey for the Eurasian otters are potentially interesting, but also raise criticisms concerning about the main topic of this manuscript based on small sampling size of the study. I agree with their concerns, and major re-formulation of the study aim will be required for this manuscript. Therefore, my editorial decision is major revision. And the referees provide excellent suggestions, which should all be addressed.

Response: The authors thank the encouragement of resubmission of our revised manuscript by the Editor. We found the reviewers’ comments highly constructive to improve the quality of our manuscript. In our revised manuscript, we have made the following three major changes: 

1) We focused more on our blocking oligonucleotide OBS1. Please see our responses to the Reviewer #1’s 1st and 2nd comments and the Reviewer #2’s 2nd comment below. 

2) We included the results of our additional experiments to assess the performance of OBS1. Please see our response to the Reviewer #2’s 2nd and 11th comments below. 

3) We re-calculated relative abundances of diet compositions since the Reviewer #1 concerned about the way we normalized our results (the Reviewer #1’s10th comment). 

Reviewer #1:

#1. This manuscript reports a study in which researchers used multiple universal metabarcoding primers to characterize the diets of Eurasian otter (Lutra lutra) and developed novel blocking primers to reduce the amount of sequencing effort lost on sequencing non-target predator DNA. The authors did a very good job succinctly outlining the importance and rationale for this research, and the methods are thoroughly and clearly outlined. However, the study in this manuscript is limited in scope, mainly serving as a proof of concept for the methodology presented, with the blocking primers for Lutra lutra being the main novel contribution this manuscript makes to the literature. The sample size is too small and the study was not designed to make any ecological comparisons, so I would recommend the authors rewrite parts of this paper to emphasize their testing of the methods and spend less time in the manuscript on trying to draw ecological conclusions from their limited data. Additionally, descriptive statistics that were reported may need to be recalculated based on rarified sequence data and I recommend more rigorous statistical testing to determine whether the sample size is adequate to draw any ecological conclusions from the data presented in this manuscript. Specific criticisms are listed below.

Response: The authors thank high-quality comments and constructive suggestions by the Reviewer #1. 

Regarding the sample size, we agree that it is a limitation of this study. As the Reviewer #1 suggested, we created the accumulation curves of unique prey taxa observed in each sample (S4 Fig). It appears that the curves do not reach to the asymptotes, suggesting that more samples are needed to capture the entire picture of diet compositions. This limitation was acknowledge in our revised manuscript. For details, please see our response to the Reviewer #1’s 12th comment below.

Due to limitation of our small sample size, we put more emphasis on our development of the blocking oligonucleotide OBS1. We included the results of our additional experiments to validate the performance of OBS1. Please see our response to the Reviewer #2’s 11th comment below. 

Regarding the rarefication of our sequencing results, we solved this problem by calculating relative abundances first, and then selecting taxa only if they were detected with more than 1% of relative abundance (Figs. 5 and 7). This approach is mathematically equivalent to the rarefaction approach in a way that taxa with less than 10 sequences were excluded from each of the libraries that were rarefied to 1,000 sequence reads per library. For details, please see our response to the Reviewer #1’s 10th comment below. 

Regarding statistical testing, we added a sub-section of statistical methods in our revised manuscript (Page 10 Lines 173-185). Statistical testing has been performed to validate the performance of our blocking oligonucleotide OBS1. The results can be found on Page 11 Line 199 to Page 12 Line 223 of our revised manuscript. 

#2. Title

I recommend reworking the title to focus more on the methodology used (mention the blocking primers)

Response: The title has been changed as follows: 

“DNA metabarcoding-based diet survey for the Eurasian otters (Lutra lutra): Development of a Eurasian otter-specific blocking oligonucleotide for 12S rDNA sequencing for vertebrates” 

#3. Abstract

12S and 16S barcoding regions are typically classified as ribosomal RNA (rRNA), not mtDNA. This should be changed throughout the manuscript.

Response: The authors believe that our targets are mitochondrially-encoded ribosomal DNA ([15, 17]). However, the term “mtDNA” was replaced by “rDNA” since “rDNA” seems more commonly used in the literature (e.g., refs [11, 17]). 

#4. Introduction

Line 37 – Citation for the sentence about the limited resolving power of morphological methods?

Response: The following citation has been added: 

Reference

9. Carss DN. 1995. Foraging behaviour and feeding ecology of the otter Lutra lutra: a selective review. Hystrix 7(1–2): 179-194

#5. Materials and methods

Line 65 – S1 Fig not included in supplementary materials

Response: S1 Fig. was uploaded to the submission system. The authors believe that it is downloadable by the Reviewers. 

#6. Lines 84-87 – This section is confusingly worded. If I understand correctly, there were three subsamples taken from each homogenized spraint, DNA was extracted from each subsample and then recombined for further analysis?

Response: Yes, the Reviewer #1’s understanding is correct. The wording has been changed for clarity as follows: 

“For the 12S and 16S rDNA analyses, three different subsamples (each with 0.2 g) of each homogenized sample in the 50 ml tube were transferred into three different 2 ml tubes for DNA extraction and purification, and recombined for elution into a tube by 50 μl of TE.” Page 5 Lines 84-88

#7. Lines 87-88 – Why was a different set of DNA used for the COI analysis?

Response: It is because experiments were conducted separately between COI and 12S/16S rDNA regions. The COI sequencing was preliminarily conducted, followed by the 12S (and 16S) rDNA sequencing. No extract was left by the preliminary COI sequencing, and we had to re-extract from the same samples for 12S/16S rDNA sequencing. The authors believe it is not critical because the results were generally congruent in terms of types of fishes detected between the COI and 12S rDNA analyses (Fig. 8 and S9 Table) even with a different set of DNA extracts from each identical spraint. 

#8. Line 90 – S2 Fig not included in supplementary materials

Response: S2 Fig. was uploaded to the submission system. The authors believe that it is downloadable by the Reviewers. 

#9. Line 113 – What was the concentration/amount in ng of the template DNA used in the PCR reactions?

Response: The information was provided in S1 Table. 

#10. Line 144 – Were sequences rarefied by sample prior to analysis? If not, this could bias the calculations of relative abundance toward the observations in samples that simply sequenced better. Taking an equal-sized random subsample of sequences from each sample after erroneous reads have been filtered out would prevent this bias (e.g. subsample 68,053 reads from each 12S sample)

Response: The authors believe that the values of relative abundances are unchanged regardless of sequencing depths under the assumption of random sampling of sequence reads (e.g., 1% for 1 out of 100 reads sequenced vs. 1% for 10 out of 1,000 reads sequenced). However, the authors agree with the Reviewer #1 that sequencing depth can affect definition of rare sequences. In our previous manuscript, we excluded unique sequences with less than 500 reads as rare sequences regardless of the variability in sequence depths across the libraries. However, this approach might disproportionately exclude rare sequences, even though they are not really rare, from libraries with the small number of sequenced reads (e.g., O1 for 16S rDNA with 1,382 reads). To circumvent this possible bias, in our present manuscript, we first calculated relative abundances for each library, and then selected taxa if they were detected with more than 1% of relative abundance (Figs. 5 and 7). This 1% threshold is equivalent of 13 reads for the library O3 for 16S rDNA with 1,382 reads, for example. In this way, taxa are not excluded, regardless of sequencing depths, if they were detected with more than 1% of relative abundance. Meanwhile, from the libraries sequenced better (e.g., O7 for 16S rDNA with 128,982 reads), taxa were conservatively excluded, even though they were detected with the large number of sequence reads (i.e., >13 reads), if they were detected below 1% of relative abundance. This approach is mathematically equivalent to the rarefaction approach in a way that taxa with less than 10 sequences were excluded from each of the libraries that were rarefied to 1,000 sequence reads per library. 

The rarefaction curves are provided in S3 Fig. The curves appear to reach asymptotes for most libraries, but some libraries (e.g., O1 for 16S rDNA) show that their curves do not reach asymptotes, suggesting that rare taxa were unlikely detected from those libraries. Thus, we decided not to discuss taxa with less than 1% of relative abundance from the discussion section and excluded from Figs. 5 and 7. 

#11. Results

Line 162 – More information could be provided on the predator sequences recovered (i.e. Were they detected in multiple samples?)

Response: The information was provided in Table 3. 

#12. 6-7 samples seems like a small sample size to claim to be a representative sample of the diets of this population. Accumulation curves of unique prey taxa observed with each sample would be helpful to visualize whether or not these samples are representative of the diet diversity of this population and could be used to help estimate what sample size would be necessary to get a representative sample in future studies. 

Response: The accumulation curves are provided in S4 Fig. It appears that the curves do not reach to the asymptotes, suggesting that more samples are needed to capture the diet compositions. In the revised manuscript, the limitation of this study was described as follows: 

“Finally, a limitation of this study was small sample size as the accumulation curves of the numbers of detected prey genera do not appear to reach to the asymptotes (S4 Fig). Future research should include more samples to capture the entire picture of the diet profiles in this study site.” Page 20 Lines 402-405

#13. Discussion

Lines 220-232 – Main results from all the analyses should be presented in the first paragraph(s). Success of the blocking primers should be one of the main takeaways. 

Response: The paragraph of the blocking oligonucleotide OBS1 was moved to the very beginning of the discussion section (Page 15 Line 304 to Page 16 Line 316). 

#14. Lines 234-266 – The discussion is hampered by the overly detailed and speculative explanations of all of the prey taxa observed in the dataset. I recommend concentrating on the most common observations, any unexpected taxa, or taxa of relevant for conservation.

Response: In our revised manuscript, taxa detected with less than 1% of relative abundance were excluded from the discussion section. Consequently, the species Tridentiger bifasciatus and Gallus gallus and their related sentences were excluded from the discussion section. Additionally, we excluded and shortened the part describing the otter’s preference of indigenous vs. exotic fishes. 

#15. Lines 247-250 – Caution should be exercised when drawing conclusions of relative biomass consumed based on relative sequence abundance in complex DNA mixtures. Due to primer and sequencing bias, the relationship between biomass and sequence abundance is often weakly correlated and cannot be assumed without test mixtures of target DNA with known biomass contributions.

Response: We are in agreement with the Reviewer #1’s comment and added the following sentence in the discussion section of our revised manuscript. 

“Moreover, caution must be taken when drawing conclusions of relative biomass consumed based on relative sequence abundance in environmental DNA since the relationship between sequence abundance and biomass often varies and weekly correlated due to differences in amplification efficiency of primers particularly in COI gene [53-55]. Additionally, a semiquantitative estimate of diet composition could be obtained in future studies after applying the robust filtering procedure described by Corse et al. [51].” Page 19 Line 395 to Page 20 Line 400

References

51. Corse E, Meglécz E, Archambaud G, Ardisson M, Martin J-F, Tougard C, et al. A from-benchtop-to-desktop workflow for validating HTS data and for taxonomic identification in diet metabarcoding studies. Mol Ecol Resour. 2017;17(6):e146–e159. pmid:28776936.

53. Bowles E, Schulte PM, Tollit DJ, Deagle BE, Trites AW. Proportion of prey consumed can be determined from faecal DNA using real-time PCR. Mol Ecol Resour. 2011;11(3):530–540. pmid:21481211.

54. Elbrecht V, Leese F. Can DNA-based ecosystem assessments quantify species abundance? Testing primer bias and biomass—sequence relationships with an innovative metabarcoding protocol. PLOS ONE. 2015;10(7):e0130324. pmid:26154168.

55. Thomas AC, Deagle BE, Eveson JP, Harsch CH, Trites AW. Quantitative DNA metabarcoding: improved estimates of species proportional biomass using correction factors derived from control material. Mol Ecol Resour. 2016;16(3):714–726. pmid:26602877.

#16. Lines 265-266 – This is somewhat speculative to assume this observation is a predation event. Could be contamination?

Response: The sentence regarding Gallus gallus have been excluded. Please see our response to the Reviewer #1’s 14th comment above. 

#17. Lines 280-281 – Are there alternative explanations to predation in this case? Would members of this family utilize fecal resources for water, food, breeding, etc.?

Response: As far as we conducted the literature survey, no such evidences could be found. It might be possible that the Reduviidae species co-inhabit with the Eurasian otters since some species are hematophagous and might feed on blood of the Eurasian otters. In our case, however, it is more natural to speculate that the Eurasian otter fed on the Reduviidae species since it was found in the spraint. 

#18. Line 287 – Incidental ingestion is also a possible explanation

Response: We agree. The following sentence has been added:

“It is also possible that they were incidentally ingested by the Eurasian otters.” Page 19 Lines 386-387

#19. Lines 302-307 – A broader discussion of the advantages and limitations of using multiple universal primer sets would be useful.

Response: Regarding the advantages and limitations of our multi locus approach, please see our responses to the Reviewer #2’s 3rd and 4th comments below. 

#20. Figures

Figure 1 – Main map needs a compass and scale bar. It is not clear to a non-Korean audience that the inset map is depicting South Korea without more geographical context. The extent indicator on the inset map does not line up with the extent of the main map. Labeling of water bodies on the main map would be helpful.

Response: Fig. 1 was re-created to provide a compass and scale bar with more details of some geographic landmarks such as cities, sea names, and the area of study site. 

Reviewer #2:

#1. In this study authors aims to investigate diet of Eurasian otters, using metabarcoding of Lutra spraints (n=7) from brackish environment in Korea. They targeted three different genes to detect the entire range diversity of prey (vertebrates and invertebrates). Authors also developed a blocking primer targeting predator DNA. Results showed that one teleost species (i.e., Carrassius auratus) is the main prey of the predator.

Response: The authors thank high-quality comments, especially regarding the methods related to the blocking oligonucleotide, by the Reviewer #2. 

#2. In this state, the ms may be criticizable by its sampling size. So, I think that authors should reformulate the aim of the study (“Investigate diet profiles of Eurasian otters”) For me it is more a feasibility study/proof of concept, and analysis and discussion should be in that direction, and lesser to trophic ecology discussion; although I agree with authors, there is an important biological result: Lutra seems to fed mostly on one species: Carassius auratus.

Here are some example of problematics that should be discuss in this sense:

Response: We agree with the Reviewer #2 that the small sample size is a limitation in this study. The Editor and the Reviewer #1 showed the same concern. They recommended to reformulate the study’s objective by putting more emphasis on our blocking oligonucleotide OBS1. We agree. In our revised manuscript, we put more emphasis on our blocking oligonucleotide OBS1. For this, we included the results of our additional experiments to assess the performance of OBS1. For details, please see our response to the Reviewer #2’s 11th comment below. 

#3. What do you advice for future studies? Is the use of multi locus approach is relevant? Because there is a trade-off between the number of amplicons (that increase the financial cost) and the complementarity between them. In your case the three locus bring different information because they were designed to target different taxa, and your result is congruent with your expectations. It should be highlighted in the discussion. Also, is the 16S amplicon that targeted the invertebrates is relevant for future studies of Lutra diet?

Response: The authors would like to advise for future studies to use the 12S rDNA sequencing analysis for vertebrate detection, including fishes, with the optional fish-specific COI gene sequencing since the memberships of detected fish genera were congruent between the two different loci (see our response to the Reviewer #2’s 4th comment below) and the broader taxonomic coverage is achieved by the 12S rDNA sequencing. The invertebrate-specific 16S rDNA sequencing is still needed to cover a broad range of the otter’s diets, such as crabs (Helicana spp.) and shrimps (Palaemon spp.). To clarify, the following sentences have been added.

“Our multi locus approach showed that the combination of 12S and 16S rDNA sequencing could cover a broad range of invertebrate and vertebrate prey animals of the Eurasian otters, including fishes detected by the COI gene sequencing. Our results suggest that the 12S and 16S rDNA sequencing analyses suffice for diet analyses of the Eurasian otters, with the optional COI gene sequencing for fish-specific detection and identification.” Page 20 Lines 412-416 

#4. Also, the challenge when using different locus in metabarcoding in my sense is related to the comparison of results. Are results of 12S and COI dataset are similar, both in term of taxa identification (qualitative aspect) and relative abundance (quantitative aspect)? Incongruent? It is relevant to keep both of them for future studies? Also, is it possible with your protocol to compare the relative importance between invertebrates and vertebrates in the diet of Lutra?

Response: We conducted a statistical test to see whether the fish genera detected by the COI sequencing were similar to those by the 12S rDNA sequencing (Page 10 Lines 181-185). We found that the variabilities in the memberships of the detected fish genera were smaller across the sequencing targets (i.e., 12S rDNA vs. COI) than across the libraries (p<0.05; PERMANOVA test) (Fig. 8), indicating that the detected fish genera are in agreement between 12S rDNA and COI sequencing analyses (Page 15 Lines 289-295). Thus, we can conclude that the results were qualitatively reproducible between the two different loci. Meanwhile, the results were not reproducible quantitatively, so that a limitation of reporting relative sequence abundances was acknowledged on Page 19 Line 395 to Page 20 Line 400 of our revised manuscript. 

Because of the congruency in the memberships of detected fish genera between the two different loci and the broader taxonomic coverage by the 12S rDNA sequencing analysis, we would like to advise for future studies to select the 12S rDNA sequencing analysis for detection of vertebrates, including fishes. To clarify, the following sentences have been added in our revised manuscript. 

“Our multi locus approach showed that the combination of 12S and 16S rDNA sequencing could cover a broad range of invertebrate and vertebrate prey animals of the Eurasian otters, including fishes detected by the COI gene sequencing. Our results suggest that the 12S and 16S rDNA sequencing analyses suffice for diet analyses of the Eurasian otters, with the optional COI gene sequencing for fish-specific detection and identification.” Page 20 Lines 412-416

It is impossible to compare the relative importance between invertebrates and vertebrates based on our multiple locus approach. This limitation has been clarified in the revised version of our manuscript. 

“It is also impossible to compare the relative importance between invertebrates and vertebrates based on our multiple locus approach.” Page 19 Lines 386-387

#5. Are the three genes enough informative to study the diet of Lutra? For example, as fishes are key prey for Lutra: Is the 12S and COI gene are sufficiently discriminant between sister species to allow identification of teleost at species level?

Response: First, the 12S rRNA gene is known to contain sufficient information to identify most of the fishes including teleost at species level [31-33]. The COI gene has a similar taxonomic resolution but has a much larger database than 12S rRNA gene. Thus, we believe that these markers are suitable for studying the diet of Lutra lutra. 

“Previous studies [31-33] demonstrated that 12S rDNA sequences provide accurate identification of most of the fishes including teleost at species level, highlighting the 12S rRNA gene as an ideal marker for diet analyses of the Eurasian otters.” Page 16 Lines 320-322

Second, the query sequence was assigned to the least common ancestor in case identification was ambiguous at the species level. This can prevent from possible misidentification at the lower taxonomic levels, and we reported only species that were unambiguously identified. 

“The ecotag command assigned the query sequence to the least common ancestor in case identification was ambiguous among sibling taxa present in the databases.” Page 9 Lines 164-165

References

31. Miya M, Sato Y, Fukunaga T, Sado T, Poulsen JY, Sato K, et al. MiFish, a set of universal PCR primers for metabarcoding environmental DNA from fishes: detection of more than 230 subtropical marine species. Royal Soc Open Sci. 2(7):150088. pmid:26587265.

32. Stoeckle MY, Soboleva L, Charlop-Powers Z. Aquatic environmental DNA detects seasonal fish abundance and habitat preference in an urban estuary. PLOS ONE. 2017;12(4):e0175186. pmid:28403183.

33. Valentini A, Taberlet P, Miaud C, Civade R, Herder J, Thomsen PF, et al. Next-generation monitoring of aquatic biodiversity using environmental DNA metabarcoding. Mol Ecol. 2016;25(4):929–942. pmid:26479867.

#6. Are the reference sequence databank are sufficiently documented for the study area to allow a more important diet study of Lutra? Is a special barcoding effort of teleost (that seems to be the preferential prey of Lutra) species is necessary?

Response: We checked the taxonomic coverages in our databases. In S6-S8 Tables, the species that were detected in this study and their sibling species listed in the databases are shown. As described in our response to the Reviewer #2’s 5th comment above, the sequence was assigned to the higher taxonomic levels such as genus or family levels in case identification was ambiguous at the species level. This means that the taxa reported at the species level were unambiguously identified among the species listed in S6-S8 Tables. 

We checked the list of prey taxa detected in spraints of Lutra lutra from Korea in a previous study [10], as well as also looked at the distribution of teleost species in Korea [34]. We confirmed the agreements with these previous studies and therefore believe that there is no significant omission of important taxa in our databases. 

“Additionally, we confirmed that fish species identified in this study are in agreements with fish species detected from spraints by a previous study [10] and spatial distributions of their habitats in Korea [34].” Page 16 Lines 323-325

References

10. Hong S, Gim J-S, Kim HG, Cowan P, Joo G-J. A molecular approach to identifying the relationship between resource use and availability in Eurasian otters (Lutra lutra). Can J Zool. 2019;97:797–804.

34. Yoon J-D, Kim J-H, Byeon M-S, Yang H-J, Park J-Y, Shim J-H, et al. Distribution patterns of fish communities with respect to environmental gradients in Korean streams. Ann Limnol - Int J Lim. 2011;47:S63–S71.

#7. Is the use of blocking primer is efficiency and does not biased the result? See my remark below.

Response: We included the results of our additional experiments to validate the performance of OBS1. For details, please see our response to the Reviewer #2’s 11th comment below. 

#8. I have also some remark on Methodologies:

I did not understood why you did not merged forward and reverse sequencing for the COI dataset? Because of potential sequencing bias (e.g., forward primer should sequenced better taxa and reverse primer sequenced better other taxa)? In general, forward/reverse primer sequencing is used when target amplicon is large. But in this ms, the two data set were not merged, and were not compared. You have to explain why you did not. 

Response: We decided not to join forward and reverse reads for COI gene sequences since the amplicon length (631 bp) was too long to be joined by 2×300 bp paired-end sequencing of Illumina Miseq. In the revised manuscript, the following explanation was added: 

“For the COI gene sequence analyses, the forward and reverse reads were not joined since the amplicon length (631 bp) was too long to be joined by 2×300 bp paired-end sequencing of Illumina Miseq. They were analyzed separately in mothur v1.40.5 [23].” Page 9 Lines 165-168

#.9 I have general remark and question on Material & Methods:

1- Why did you not use replicates sample?

Response: We agree that the inclusion of biological replicates is important to define measurement precisions of given analytical systems. Unfortunately, we failed to include them for the analyses of spraint samples collected in this study. We believe, however, that the lack of biological replicates does not affect our conclusions of the validity of our blocking oligonucleotide and the overall tendencies of diet profiles observed across the seven different spraints collected and characterized by the sequencing analyses targeting the three different loci in this study. Nonetheless, we agree with its importance. We will keep in mind for our future investigations. 

#10. Relative abundance of reads: if reads proportion is used for ribosomal genes in metabarcoding, it is more controversial when using COI gene. You should evoke this in your discussion and explain how you deal with your data in this context.

Response: We agree with reviewer’s concern about using reads proportion for COI gene, hence we added the following caution statement in discussion. 

“Moreover, caution must be taken when drawing conclusions of relative biomass consumed based on relative sequence abundance in environmental DNA since the relationship between sequence abundance and biomass often varies and weekly correlated due to differences in amplification efficiency of primers particularly in COI gene [53-55]. Additionally, a semiquantitative estimate of diet composition could be obtained in future studies after applying the robust filtering procedure described by Corse et al. [51].” Page 19 Line 385 to Page 20 Line 400

#11. The use of blocking primer is useful, but difficult to design. In my experience, its specificity should always be check by in silico tests (as you did) but also by in vivo tests. For that, you should use positive control, like a mix of teleost and invertebrates DNA (target prey) and a negative control with only DNA targeted by the blocking primer (here the lutra lutra DNA), in order to ensure 1) that prey DNA is not blocked, and 2) to assess the efficiency of the blocking primer to inhibit the amplification of the predator DNA. Indeed, maybe you found few predator DNA just because the PCR primer did not work well on predator DNA. In this case, blocking primer is not essential and should be removed to avoid potential interaction with prey DNA. The validation of blocking primer need both approach.

Response: After submission of the previous version of our manuscript, we conducted two additional experiments to assess the performance of our blocking oligonucleotide, since we noticed the issues pointed out by the Reviewer #2. First, we conducted the negative control experiment to confirm the inhibition of amplification of the synthesized otter’s DNA by our blocking oligonucleotide (Page 7 Lines108-120). The experiment shows the amplification of otter’s DNA was successfully inhibited by our blocking oligonucleotide OBS1 (Fig. 2). Second, we conducted the additional sequencing experiment without the blocking oligonucleotide OBS1 (raw sequence data are available under the accession number of PRJNA565647 of NCBI), and compared the taxonomic compositions with those sequenced with the blocking oligonucleotide OBS1 (Page 8 Lines 132-314, Page 11 Lines 199-208). We successfully confirmed that the variabilities in unique sequence structures were smaller across the library preparation methods (i.e., with vs. without OBS1) than across the samples (p<0.001; PERMANOVA) (Fig. 3), indicating little inhibitory effects on amplification of prey DNA by the blocking oligonucleotide OBS1 during library preparation. 

#12. See more on the interest to use positive and negative control in diet metabarcoding in Corse, E. et al. A from-benchtop-to-desktop workflow for validating HTS data and for taxonomic identification in diet metabarcoding studies. Mol Ecol Resour 17, e146–e159 (2017).

Response: The authors thank the Reviewer #2 for providing this invaluable source of information regarding diet analyses by DNA metabarcoding. We will keep in mind for our future studies of diet analyses by DNA metabarcoding. 

As described in our response to the Reviewer #2’s 11th comment above, we conducted the negative control experiment to confirm the inhibition of amplification of the Eurasian otter’s DNA by our blocking oligonucleotide OBS1. Although we did not conduct the positive control experiment using a mock community of prey taxa, we conducted the additional sequencing analysis without the blocking oligonucleotide OBS1, and compared the taxonomic compositions with those sequenced with the blocking oligonucleotide OBS1 (Page 8 Lines 132-314, Page 11 Lines 199-208). We successfully confirmed that the variabilities in unique sequence structures were smaller across the library preparation methods (i.e., with vs. without OBS1) than across the samples (p<0.001; PERMANOVA) (Fig. 3), indicating little inhibitory effects on amplification of prey DNA by the blocking oligonucleotide OBS1 during library preparation. With these two additional experiments, we could conclude that: 1) prey DNA was not significantly blocked; and 2) the predator DNA was blocked by our blocking oligonucleotide OBS1. 

#13. Minor remark

l.220 I think that you should moderate your conclusion. Indeed, are you sure that your PCR primer were not simply better for fish than for vertebrates? Is your blocking primer should inhibit some vertebrates DNA amplification?

Response: The word “revealed” has been changed to the word “showed”. We believe that there is no significant omissions of prey taxa by our designed blocking oligonucleotide OBS1 since we confirmed little inhibitory effects on prey DNA by OBS1 during library preparation (Fig. 3). 

#14. l.255 Some terms should be check to avoid confusion: What do you mean by “fish susceptibility to predation” ? Prey selection? Also, what do you mean by relative abundance? Their relative abundance in the environment?

Response: This part was deleted from the revised version of our manuscript because of the suggestion made by the Reviewer #1. Please see our response to the Reviewer #1’s 14 comment above. 

#15. l.289/290 Not totally true. It’s possible to detect cannibalism if you use “a variant-centered metabarcoding data filtering”, rather than a “sequence group” filtration. When using COI, such filtering process allow to obtain the different haplotype of the same species; see in Corse et al., 2017. Using this approach, we detected intra species predation on trutta and Gobiidae, see Corse et al., 2019:

Corse, E. et al. One-locus-several-primers: A strategy to improve the taxonomic and haplotypic coverage in diet metabarcoding studies. Ecology and Evolution 9, 4603–4620 (2019).

Response: Thanks for the information. This is really intriguing. The sentence has been revised as follows:

“Other potential limitations include their inability of detecting possible cannibalism [46] because, unless the specialized approaches, such as the so-called the variant-centered approach for haplotype detection [47, 48], are used, DNA metabarcoding cannot distinguish DNA fragments from cannibalized and cannibalizing individuals.” Page 19 Lines 387-391

#16. l.291-293 This process named secondary predation.

Response: We thank the Reviewer #2 for teaching us the terminology. The term was used as follows: 

“Additionally, the organisms detected in spraints might be due to the secondary predation.” Page 19 Lines 391-392

#17. l.296 I think you had to precise what type of blocking primer. Indeed, there is not only one type of blocking primer.

Response: The type of blocking oligonucleotides used in this study was with a 3-carbon spacer at the 3’-end. We have added this detail in the revised version of our manuscript. 

“The designed oligonucleotide OBS1 with a 3-carbon spacer at the 3’-end (Table 1) specifically binds to and blocks amplification of the12S rDNA region of the Eurasian otters.” Page 6 Lines 97-99

#18. l.299-302 In my sense, you cannot be sure with your data that it is your blocking primer that inhibit predator DNA amplification (see my general remark).

Response: Please see our response to the Reviewer #2’s 11th comment. 

#19. l.303 Your argument is good only if the COI dataset is representative of the fishes diversity in the diet of Lutra. So, are you sure that your COI primer allow to obtain a representative picture of the diversity of fishes in the diet? In fact, you have to discuss that your COI primer set allow to amplify all potentially fishes eaten by Lutra.

Response: We agree. We cannot know about the inhibitory effects of OBS1 by the comparison with the COI results since we do not really know about the selectivity of the COI primers. The part has been deleted from our revised manuscript. Instead, we added a sentence describing the little inhibitory effects based on the comparison with the 12S rDNA sequencing analysis without OBS1 (Page 8 Lines 132-314, Page 11 Lines 199-208). 

#20. l.317 Please, check this: “…however [3, 4]”

Response: The sentence has been fixed. 

#21. Figures

Fig.1 Information are lacking. Provides some geographic landmarks such as cities, river and ocean names, area of study site, map scales.

Response: The suggested changes have been made in the revised Fig. 1. 

#22. Fig.2 This figure is hard to follow.

1) I think you have to choose, giving relative abundance in the text or in the figure. For me, relative abundance by taxa is redundant in a pie chart figure. Keep % only in the text.

Response: The authors wish to keep relative abundance values both in the figure and text since figures have to be self-contained, generally speaking. 

#23. 2) You should remove color, or keep unique color code for the four pie chart.

Response: The unique color code has been used for the 12S rDNA and COI data (Fig. 4) (i.e., the same color is used for the same taxon). The 16S rDNA data are separated into a new figure (Fig. 6) since the taxa are different and the same color code cannot be used for them. 

#24. 3) You should indicate in a subtitle above each pie chart the data used. Example for a) 12S gene sequence b) 16S gene sequence ….

Response: The suggested change has been made. 

#25. Fig.3 As for Fig 2 you should add subtitle. Also you can remove phylogenetical trees because they did not supply any crucial information and did not revealed true phylogenetical relationship. 

Response: The suggested changes have been made.

---

## [Decision Letter · Decision Letter 1]

12 Nov 2019

PONE-D-19-19235R1

DNA metabarcoding-based diet survey for the Eurasian otters (Lutra lutra): Development of a Eurasian otter-specific blocking oligonucleotide for 12S rDNA sequencing for vertebrates

PLOS ONE

Dear Dr. Yamamoto,

Thank you for submitting your manuscript to PLOS ONE. After careful consideration, we feel that it has merit but does not fully meet PLOS ONE’s publication criteria as it currently stands. Therefore, we invite you to submit a revised version of the manuscript that addresses the points raised during the review process.

The manuscript has been greatly improved. However, you will see that some questions still exist. I invite you to submit a revised version of your manuscript according to the reviewer's comments. Then I can make a final decision.

We would appreciate receiving your revised manuscript by Dec 27 2019 11:59PM. To enhance the reproducibility of your results, we recommend that if applicable you deposit your laboratory protocols in protocols.io, where a protocol can be assigned its own identifier (DOI) such that it can be cited independently in the future. For instructions see: http://journals.plos.org/plosone/s/submission-guidelines#loc-laboratory-protocols

We look forward to receiving your revised manuscript.

Kind regards,

Zuogang Peng, Ph.D.

Academic Editor

PLOS ONE

Reviewers' comments:

Reviewer's Responses to Questions

**Comments to the Author**

1. If the authors have adequately addressed your comments raised in a previous round of review and you feel that this manuscript is now acceptable for publication, you may indicate that here to bypass the “Comments to the Author” section, enter your conflict of interest statement in the “Confidential to Editor” section, and submit your "Accept" recommendation.

Reviewer #1: All comments have been addressed

Reviewer #2: All comments have been addressed

2. Is the manuscript technically sound, and do the data support the conclusions?

Reviewer #1: Yes

Reviewer #2: Yes

3. Has the statistical analysis been performed appropriately and rigorously? 

Reviewer #1: Yes

Reviewer #2: No

4. Have the authors made all data underlying the findings in their manuscript fully available?

Reviewer #1: Yes

Reviewer #2: Yes

5. Is the manuscript presented in an intelligible fashion and written in standard English?

Reviewer #1: Yes

Reviewer #2: Yes

6. Review Comments to the Author

Reviewer #1: The authors have addressed all of my major concerns with the original manuscript. I think the shift to focus on the development and testing of the blocking primers has given the manuscript a clearer purpose and strengthened the results. Additionally, the authors have more adequately addressed the limitations raised during the previous review. I have minor grammatical and clarification suggestions outlined below.

Throughout manuscript: 12S and 16S sequences were referred to as "rDNA", this should be "rRNA".

Line 89: remove "respectively"

Lines 304-307: The first paragraph of the discussion would be stronger if it focused on the main takeaway from this study (that the blocking primers are very effective).

Line 318: "fishes (95%) was" should be "fishes (95%) were"

Line 321: "teleost" should be "teleosts"

Reviewer #2: Dear editor,

The authors have taken into account the main remarks of the two reviewers, especially by changing the aim of their ms, but also by adding new tests and table (especially the very informative table 3) on the performance of the blocking oligonucleotide and its potential risk of inhibiting non target DNA amplification. Overall, the ms has been significantly improve, and is for me close to a publishable state.

But I have a last concern on the statistical tests used for testing library and loci effect. Maybe I did not understand fully how the authors used the permanova, but authors should give more information of test outputs, and maybe rethink these tests and/or how they interpret it. Alternatively, I have made proposals to test (in my opinion) the words of authors:

1. To demonstrate that “variabilities in the memberships of the detected fish genera were smaller across the sequencing loci (i.e., 12S rDNA vs. COI) than across the libraries (p<0.05)”

I would compare the mean intra-sample distances (between the two loci) vs mean intra-library distances.

But I think (see my remark below) that it will be better to compare in fact these three distances:

Intra-samples distances (between loci) vs inter-samples distances using COI vs inter-samples distances using 12S

2. To demonstrate that “we confirmed that the variabilities in unique sequence structures were smaller across the library preparation methods (i.e., with vs. without OBS1) than across the samples”

I would compare the mean between-sample distances vs mean intra-library distances.

Remark:

L. 168 You define here what you call “unique sequence”. As you will usually use this term in the rest of the ms (result part mostly), and because the sequence-centered view is not “familiar” for metabarcoding process, I propose you to quickly remember what you stipulate by this term when you use it.

L.185 Maybe PCoA should be announced at this step.

L.312-316 I am confused with your conclusions. If you tested both gene and library effect on jaccard dissimiliraty matrix, and that the result is significant, it does not obligatory indicated that gene effect is larger than library effect. For that, you may indicated their relative explained variance using the R2 if you use Adonis function of vegan package. But the output of permanova is lacking in your ms, and we don’t know to what corresponded your significant value.

Also, maybe using betadisper function that estimate the mean distance of individuals around barycenter should be serve to propose a proxy that indicate the amount of dispersion for a grouping factor. In this case, you may calculate the global dispersion among individuals for 12S, and the global dispersion for COI. These two values could be indicate in parallel to your permanova test. To illustrate this dispersion, maybe you can boxplot the jaccard pairwise-distance among specimens for each loci, I think that it should more support your words than the dendogram.

But I have a more profound remark on your reasoning. If you prove that library dispersion is higher than loci dispersion, it does not prove that result obtained using different loci are congruent. In my sens, you have to adding the specimen effect. Loci effect may be overlooked if the mean intra-sample (between loci result of a same specimen) distance is lower than the inter-sample distance obtained with each loci.

L.338-341 As it is a similar analyses, see my remark L.312-316

7. PLOS authors have the option to publish the peer review history of their article (what does this mean?). If published, this will include your full peer review and any attached files.

Reviewer #1: No

Reviewer #2: Yes: CORSE

---

## [Author Response · Author response to Decision Letter 1]

15 Nov 2019

The authors thank additional comments provided from the Reviewers #1 and #2. Below, please find our responses to the reviewers’ comments. In addition to responding to the reviewers’ comments, we made a final editing of our manuscript, especially for the abstraction and introduction sections and the last paragraph of discussion section, to improve readability. The authors believe that there is no significant content change incurred due to editing. The page numbers in our responses refer to the clean version of our revised manuscript. 

Reviewer #1:

1. The authors have addressed all of my major concerns with the original manuscript. I think the shift to focus on the development and testing of the blocking primers has given the manuscript a clearer purpose and strengthened the results. Additionally, the authors have more adequately addressed the limitations raised during the previous review. I have minor grammatical and clarification suggestions outlined below.

Response: The authors thank positive appraisal from the Reviewer #1. 

2. Throughout manuscript: 12S and 16S sequences were referred to as "rDNA", this should be "rRNA".

Response: The authors believe that the term “rDNA” can work as it represents DNA (gene) that encodes rRNA (transcripts), i.e., rRNA gene. Meanwhile, the term “rRNA” without the word “gene” does not work since we sequenced DNA, not RNA. So, the authors believe that both of the terms “rDNA” and “rRNA gene” can work while the term “rRNA” without the word “gene” does not work. As the Reviewer #1 suggested, however, the term “rDNA” has been replaced with the term “rRNA gene” throughout the manuscript since it seems that the term “rRNA gene” is more commonly used than the term “rDNA” in literature. Accordingly, the title of the paper has been also changed as follows: 

“DNA metabarcoding-based diet survey for the Eurasian otter (Lutra lutra): Development of a Eurasian otter-specific blocking oligonucleotide for 12S rRNA gene sequencing for vertebrates” 

3. Line 89: remove "respectively"

Response: Corrected. 

4. Lines 304-307: The first paragraph of the discussion would be stronger if it focused on the main takeaway from this study (that the blocking primers are very effective).

Response: To emphasize our development of the blocking oligonucleotide OBS1, a sentence has been added at the beginning of the first paragraph of the discussion section as follows: 

“In this study, we designed a Eurasian otter-specific blocking oligonucleotide (OBS1) for their diet analyses based on 12S rRNA gene sequencing for vertebrates.” Page 16 Lines 320-321

In addition, the following sentence has been included in the conclusion section.

“In particular, we developed a Eurasian otter-specific blocking oligonucleotide (OBS1) for 12S rRNA gene sequencing for vertebrates.” Page 21 Lines 424-426 

5. Line 318: "fishes (95%) was" should be "fishes (95%) were"

Response: Corrected. 

6. Line 321: "teleost" should be "teleosts"

Response: Corrected. 

Reviewer #2:

1. The authors have taken into account the main remarks of the two reviewers, especially by changing the aim of their ms, but also by adding new tests and table (especially the very informative table 3) on the performance of the blocking oligonucleotide and its potential risk of inhibiting non target DNA amplification. Overall, the ms has been significantly improve, and is for me close to a publishable state. 

Response: The authors thank positive appraisal from the Reviewer #2. 

2. But I have a last concern on the statistical tests used for testing library and loci effect. Maybe I did not understand fully how the authors used the permanova, but authors should give more information of test outputs, and maybe rethink these tests and/or how they interpret it. Alternatively, I have made proposals to test (in my opinion) the words of authors:

Response: The authors agree that the test results of PERMANOVA cannot tell what we insisted regarding the levels of intra- vs. inter-group variabilities. The additional statistical analyses were performed as the Reviewer #2 suggested. For details, please see our response below. 

3. To demonstrate that “variabilities in the memberships of the detected fish genera were smaller across the sequencing loci (i.e., 12S rDNA vs. COI) than across the libraries (p<0.05)”

I would compare the mean intra-sample distances (between the two loci) vs mean intra-library distances. But I think (see my remark below) that it will be better to compare in fact these three distances: Intra-samples distances (between loci) vs inter-samples distances using COI vs inter-samples distances using 12S

Response: The suggested analysis has been performed. Specifically, we made a boxplot showing the distributions of the pair-wise intra-sample distances obtained by 12S rRNA vs. COI gene sequencing, the inter-sample distances obtained by 12S rRNA gene sequencing, and the inter-sample distances obtained by COI gene sequencing (Fig. 8b). The intra-sample distances were smallest among other distances. In particular, they were significantly smaller than the inter-sample distances based on 12S rRNA gene sequencing. We believe that these results support our claim though they were not significantly smaller than the inter-sample distances based on COI gene sequencing. To describe these statistical results, the following sentences have been added: 

“Fig. 8 shows that the variabilities in the memberships of detected fish genera were smaller across the sequencing loci (i.e., 12S rRNA gene vs. COI) than across the samples, with a statistical significance observed between the intra-sample distances based on the two different loci and the inter-sample distances based on 12S rRNA gene sequencing, indicating that the detected fish genera are in agreement between the two locus analyses in terms of detected fish memberships.” Page 16 Lines 301-306 

4. To demonstrate that “we confirmed that the variabilities in unique sequence structures were smaller across the library preparation methods (i.e., with vs. without OBS1) than across the samples”

I would compare the mean between-sample distances vs mean intra-library distances.

Response: The suggested analysis has been performed by making a boxplot showing the distributions of the pair-wise intra-sample distances obtained with vs. without blocking oligonucleotides OBS1, the inter-sample distances obtained with OBS1, and the inter-sample distances obtained without OBS1 (Fig. 3b). The beta dispersion was smallest for the intra-sample group. We believe that these results support our claim. 

5. Remark:

L. 168 You define here what you call “unique sequence”. As you will usually use this term in the rest of the ms (result part mostly), and because the sequence-centered view is not “familiar” for metabarcoding process, I propose you to quickly remember what you stipulate by this term when you use it.

Response: The following explanation was added for the term “unique sequences”. 

“First, identical reads were binned to generate a set of unique sequences, each with an identical length and nucleotide sequence.” Page 9 Lines 167-168

6. L.185 Maybe PCoA should be announced at this step.

Response: The information was included in the section of statistical analyses. Additionally, we included the information of methods to analyze beta dispersion. The section has been revised as follows: 

“R version 3.6.0 was used for statistical analyses. The vegan package version 2.5-6 [28] was used to check for possible changes in the sequence structures by the blocking oligonucleotides OBS1 and for the difference in fish memberships detected by sequencing of two different loci of 12S rRNA and COI genes. For the first purpose, the Bray-Curtis dissimilarities of unique sequence structures were characterized and compared across the 12S rRNA gene libraries prepared with vs. without blocking oligonucleotides OBS1 and HomoB after excluding the sequence reads assigned to Lutra lutra and Homo sapiens. For the second purpose, the Jaccard distances were calculated to characterize the differences of memberships of fish genera detected by 12S rRNA vs. COI gene sequencing analyses. Principal coordinate analysis plot and cluster dendrogram were created based on the Bray-Curtis and Jaccard distances calculated for the first and second purposes, respectively. Permutational multivariate analysis of variance (PERMANOVA) tests were conducted and followed by Kruskal-Wallis rank sum tests with post hoc Wilcoxon rank-sum tests with Bonferroni correction to compare pair-wise intra- and inter-sample distances to assess the degrees of variabilities across the samples and across the library preparation methods. P-values less than 0.05 were considered statistically significant.” Page 10 Lines 174-188 

7. L.312-316 I am confused with your conclusions. If you tested both gene and library effect on jaccard dissimiliraty matrix, and that the result is significant, it does not obligatory indicated that gene effect is larger than library effect. For that, you may indicated their relative explained variance using the R2 if you use Adonis function of vegan package. But the output of permanova is lacking in your ms, and we don’t know to what corresponded your significant value.

Response: The authors thanks the Reviewer #1 to remind us the importance of showing the effect sizes, which are perhaps more important than p-values. Now, the effect sizes in terms of r2 were included in the revised Figs. 3 and 8. 

8. Also, maybe using betadisper function that estimate the mean distance of individuals around barycenter should be serve to propose a proxy that indicate the amount of dispersion for a grouping factor. In this case, you may calculate the global dispersion among individuals for 12S, and the global dispersion for COI. These two values could be indicate in parallel to your permanova test. To illustrate this dispersion, maybe you can boxplot the jaccard pairwise-distance among specimens for each loci, I think that it should more support your words than the dendogram.

Response: Please see our response to the Reviewer #2’s 3rd comment above. 

9. But I have a more profound remark on your reasoning. If you prove that library dispersion is higher than loci dispersion, it does not prove that result obtained using different loci are congruent. In my sense, you have to adding the specimen effect. Loci effect may be overlooked if the mean intra-sample (between loci result of a same specimen) distance is lower than the inter-sample distance obtained with each loci. 

Response: As suggested, we compared the differences in distances in three different groups, i.e., intra-sample (12S vs. COI), inter-sample (12S), and inter-sample (COI). For details, please see our response to the Reviewer #2’s 3rd comment above. 

10. L.338-341 As it is a similar analyses, see my remark L.312-316

Response: In Fig. 3, r2 was included.

---

## [Editor Report · Decision Letter 2]

25 Nov 2019

DNA metabarcoding-based diet survey for the Eurasian otter (Lutra lutra): Development of a Eurasian otter-specific blocking oligonucleotide for 12S rRNA gene sequencing for vertebrates

PONE-D-19-19235R2

Dear Dr. Yamamoto,

We are pleased to inform you that your manuscript has been judged scientifically suitable for publication and will be formally accepted for publication once it complies with all outstanding technical requirements.

With kind regards,

Zuogang Peng, Ph.D.

Academic Editor

PLOS ONE
---

## [Editor Report · Acceptance letter]

3 Dec 2019

PONE-D-19-19235R2 

DNA metabarcoding-based diet survey for the Eurasian otter (*Lutra lutra*): Development of a Eurasian otter-specific blocking oligonucleotide for 12S rRNA gene sequencing for vertebrates 

Dear Dr. Yamamoto:

I am pleased to inform you that your manuscript has been deemed suitable for publication in PLOS ONE. Congratulations! Your manuscript is now with our production department. 

With kind regards,

on behalf of

Dr. Zuogang Peng 

Academic Editor

PLOS ONE